# Binocular rivalry reveals an out-of-equilibrium neural dynamics suited for decision-making

Robin Cao[1,2,3], Alexander Pastukhov[1], Stepan Aleshin[1], Maurizio Mattia[3†], Jochen Braun[1*†]

[1]Cognitive Biology, Center for Behavioral Brain Sciences, Magdeburg, Germany; [2]Gatsby Computational Neuroscience Unit, London, United Kingdom; [3]Istituto Superiore di Sanità, Rome, Italy

**Abstract** In ambiguous or conflicting sensory situations, perception is often 'multistable' in that it perpetually changes at irregular intervals, shifting abruptly between distinct alternatives. The interval statistics of these alternations exhibits quasi-universal characteristics, suggesting a general mechanism. Using binocular rivalry, we show that many aspects of this perceptual dynamics are reproduced by a hierarchical model operating out of equilibrium. The constitutive elements of this model idealize the metastability of cortical networks. Independent elements accumulate visual evidence at one level, while groups of coupled elements compete for dominance at another level. As soon as one group dominates perception, feedback inhibition suppresses supporting evidence. Previously unreported features in the serial dependencies of perceptual alternations compellingly corroborate this mechanism. Moreover, the proposed out-of-equilibrium dynamics satisfies normative constraints of continuous decision-making. Thus, multistable perception may reflect decision-making in a volatile world: integrating evidence over space and time, choosing categorically between hypotheses, while concurrently evaluating alternatives.

**\*For correspondence:**
jochen.braun@ovgu.de

[†]These authors contributed equally to this work

**Competing interest:** The authors declare that no competing interests exist.

## Introduction

In deducing the likely physical causes of sensations, perception goes beyond the immediate sensory evidence and draws heavily on context and prior experience (*von Helmholtz, 1867*; *Barlow et al., 1972*; *Gregory, 1980*; *Rock, 1983*). Numerous illusions in visual, auditory, and tactile perception – all subjectively compelling, but objectively false – attest to this extrapolation beyond the evidence. In natural settings, perception explores alternative plausible causes of sensory evidence by active readjustment of sensors ('active perception,' *Mirza et al., 2016*; *Yang et al., 2018*; *Parr and Friston, 2017a*). In general, perception is thought to actively select plausible explanatory hypotheses, to predict the sensory evidence expected for each hypothesis from prior experience, and to compare the observed sensory evidence at multiple levels of scale or abstraction ('analysis by synthesis,' 'predictive coding,' 'hierarchical Bayesian inference,' *Yuille and Kersten, 2006*, *Rao and Ballard, 1999*, *Parr and Friston, 2017b*, *Pezzulo et al., 2018*). Active inference engages the entire hierarchy of cortical areas involved in sensory processing, including both feedforward and feedback projections (*Bar, 2009*; *Larkum, 2013*; *Shipp, 2016*; *Funamizu et al., 2016*; *Parr et al., 2019*).

The dynamics of active inference becomes experimentally observable when perceptual illusions are 'multistable' (*Leopold and Logothetis, 1999*). In numerous ambiguous or conflicting situations, phenomenal experience switches at irregular intervals between discrete alternatives, even though the sensory scene is stable (*Necker, 2009*; *Wheatstone, 1838*; *Rubin, 1958*; *Attneave, 1971*; *Ramachandran and Anstis, 2016*; *Pressnitzer and Hupe, 2006*; *Schwartz et al., 2012*). Multistable illusions are

enormously diverse, involving visibility or audibility, perceptual grouping, visual depth or motion, and many kinds of sensory scenes, from schematic to naturalistic. Average switching rates differ greatly and range over at least two orders of magnitude (*Cao et al., 2016*), depending on sensory scene, perceptual grouping (*Wertheimer, 1912*; *Koffka, 1935*; *Ternus, 1926*), continuous or intermittent presentation (*Leopold and Logothetis, 2002*; *Maier et al., 2003*), attentional condition (*Pastukhov and Braun, 2007*), individual observer (*Pastukhov et al., 2013c*; *Denham et al., 2018*; *Brascamp et al., 2019*), and many other factors.

In spite of this diversity, the stochastic properties of multistable phenomena appear to be quasi-universal, suggesting that the underlying mechanisms may be general. Firstly, average dominance duration depends in a characteristic and counterintuitive manner on the strength of dominant and suppressed evidence ('Levelt's propositions I–IV,' *Levelt, 1965*; *Brascamp et al., 2006*; *Klink et al., 2016*; *Kang, 2009*; *Brascamp et al., 2015*; *Moreno-Bote et al., 2010*). Secondly, the statistical distribution of dominance durations shows a stereotypical shape, resembling a gamma distribution with shape parameter $r \simeq 3 - 4$ ('scaling property,' *Cao et al., 2016*; *Fox and Herrmann, 1967*; *Blake et al., 1971*; *Borsellino et al., 1972*; *Walker, 1975*; *De Marco et al., 1977*; *Murata et al., 2003*; *Brascamp et al., 2005*; *Pastukhov and Braun, 2007*; *Denham et al., 2018*; *Darki and Rankin, 2021*). Thirdly, the durations of successive dominance periods are correlated positively, over at least two or three periods (*Fox and Herrmann, 1967*; *Walker, 1975*; *Van Ee, 2005*; *Denham et al., 2018*).

Here, we show that these quasi-universal characteristics are comprehensively and quantitatively reproduced, indeed guaranteed, by an interacting hierarchy of birth-death processes operating out of equilibrium. While the proposed mechanism combines some of the key features of previous models, it far surpasses their explanatory power.

Several possible mechanisms have been proposed for perceptual dominance, the triggering of reversals, and the stochastic timing of reversals. That a single, coherent interpretation typically dominates phenomenal experience is thought to reflect competition (explicit or implicit) at the level of explanatory hypotheses (e.g., *Dayan, 1998*), sensory inputs (e.g., *Lehky, 1988*), or both (e.g., *Wilson, 2003*). That a dominant interpretation is occasionally supplanted by a distinct alternative has been attributed to fatigue processes (e.g., neural adaptation, synaptic depression, *Laing and Chow, 2002*), spontaneous fluctuations ('noise,' e.g., *Wilson, 2007*, *Kim et al., 2006*), stochastic sampling (e.g., *Schrater and Sundareswara, 2006*), or combinations of these (e.g., adaptation and noise, *Shpiro et al., 2009*; *Seely and Chow, 2011*; *Pastukhov et al., 2013c*). The characteristic stochasticity (gamma-like distribution) of dominance durations has been attributed to Poisson counting processes (e.g., birth-death processes, *Taylor and Ladridge, 1974*; *Gigante et al., 2009*; *Cao et al., 2016*) or stochastic accumulation of discrete samples (*Murata et al., 2003*; *Schrater and Sundareswara, 2006*; *Sundareswara and Schrater, 2008*; *Weilnhammer et al., 2017*).

'Dynamical' models combining competition, adaptation, and noise capture well the characteristic dependence of dominance durations on input strength ('Levelt's propositions') (*Laing and Chow, 2002*; *Wilson, 2007*; *Ashwin and Aureliu, 2010*), especially when inputs are normalized (*Moreno-Bote et al., 2007*; *Moreno-Bote et al., 2010*; *Cohen et al., 2019*), and when the dynamics emphasize noise (*Shpiro et al., 2009*; *Seely and Chow, 2011*; *Pastukhov et al., 2013c*). However, such models do not preserve distribution shape over the full range of input strengths (*Cao et al., 2016*; *Cohen et al., 2019*). On the other hand, 'sampling' models based on discrete random processes preserve distribution shape (*Taylor and Ladridge, 1974*; *Murata et al., 2003*; *Schrater and Sundareswara, 2006*; *Sundareswara and Schrater, 2008*; *Cao et al., 2016*; *Weilnhammer et al., 2017*), but fail to reproduce the dependence on input strength. Neither type of model accounts for the sequential dependence of dominance durations (*Laing and Chow, 2002*).

Here, we reconcile 'dynamical' and 'sampling' approaches to multistable perception, extending an earlier effort (*Gigante et al., 2009*). Importantly, every part of the proposed mechanism appears to be justified normatively in that it may serve to optimize perceptual choices in a general behavioral situation, namely, continuous inference in uncertain and volatile environments (*Bogacz, 2007*; *Veliz-Cuba et al., 2016*). We propose that sensory inputs are represented by birth-death processes in order to accumulate sensory information over time and in a format suited for Bayesian inference (*Ma et al., 2006*; *Pouget et al., 2013*). Further, we suggest that explanatory hypotheses are evaluated competitively, with a hypothesis attaining dominance (over phenomenal experience) when its support exceeds the alternatives by a certain finite amount, consistent with optimal decision-making between

multiple alternatives (*Bogacz, 2007*). Finally, we assume that a dominant hypothesis suppresses its supporting evidence, as required by 'predictive coding' implementations of hierarchical Bayesian inference (*Pearl, 1988*; *Rao and Ballard, 1999*; *Hohwy et al., 2008*). In contrast to many previous models, we do not require a local mechanisms of fatigue, adaptation, or decay.

Based on these assumptions, the proposed mechanism reproduces dependence on input strength, as well as distribution of dominance durations and positive sequential dependence. Additionally, it predicts novel and unsuspected dynamical features confirmed by experiment.

## Results

Below we introduce each component of the mechanism and its possible normative justification, before describing out-of-equilibrium dynamics resulting from the interaction of all components. Subsequently, we compare model predictions with multistable perception of human observers, specifically, the dominance statistics of binocular rivalry (BR) at various combinations of left- and right-eye contrasts (*Figure 1a*).

### Hierarchical dynamics

#### Bistable assemblies: 'local attractors'

As operative units of sensory representation, we postulate neuronal assemblies with bistable 'attractor' dynamics. Effectively, assembly activity moves in an energy landscape with two distinct quasi-stable states – dubbed 'on' and 'off' – separated by a ridge (*Figure 1b*). Driven by noise, assembly activity mostly remains near one quasi-stable state ('on' or 'off'), but occasionally 'escapes' to the other state (*Kramers, 1940*; *Hanggi et al., 1990*; *Deco and Hugues, 2012*; *Litwin-Kumar and Doiron, 2012*; *Huang and Doiron, 2017*).

An important feature of 'attractor' dynamics is that the energy of quasi-stable states depends sensitively on external input. Net positive input destabilizes (i.e., raises the potential of) the 'off' state and stabilizes (i.e., lowers the potential of) the 'on' state. Transition rates $\nu^\pm$ are even more sensitive to external input as they depend approximately exponentially on the height of the energy ridge ('activation energy').

*Figure 1b* illustrates 'attractor' dynamics for an assembly of 150 spiking neurons with activity levels of approximately $7\,Hz$ and $21\,Hz$ per neuron in the 'off' and 'on' states, respectively. Full details are provided in Appendix 1, section Metastable population dynamics, and *Appendix 1—figure 2*.

#### Binary stochastic variables

Our model is independent of neural details and relies exclusively on an idealized description of 'attractor' dynamics. Specifically, we reduce bistable assemblies to discretely stochastic, binary activity variables $x(t) \in \{0, 1\}$, which activate and inactivate with Poisson rates $\nu^+$ and $\nu^-$, respectively. These rates $\nu^\pm(s)$ vary exponentially and anti-symmetrically with increments or decrements of activation energy $\Delta u = u(s) + u^0$:

$$\nu^+ = \frac{\nu}{2} \exp\left(\frac{\Delta u}{2}\right), \qquad \nu^- = \frac{\nu}{2} \exp\left(-\frac{\Delta u}{2}\right) \tag{1}$$

where $u^0$ and $\nu$ are baseline potential and baseline rate, respectively, and where the input-dependent part $u(s) = ws$ varies linearly with input $s$, with synaptic coupling constant $w$ (see Appendix 1, section Metastable population dynamics and *Appendix 1—figure 2e*).

#### Pool of $N$ binary variables

An extended network, containing $N$ individually bistable assemblies with shared input $s$, reduces to a 'pool' of $N$ binary activity variables $x_i(t) \in \{0, 1\}$ with identical rates $\nu_\pm(s)$. Although all variables are independently stochastic, they are coupled through their shared input $s$. The number of active variables $n(t) = \sum_i x_i(t)$ or, equivalently, the active fraction $x(t) = n(t)/N$, forms a discretely stochastic process ('birth-death' or 'Ehrenfest' process; *Karlin and McGregor, 1965*).

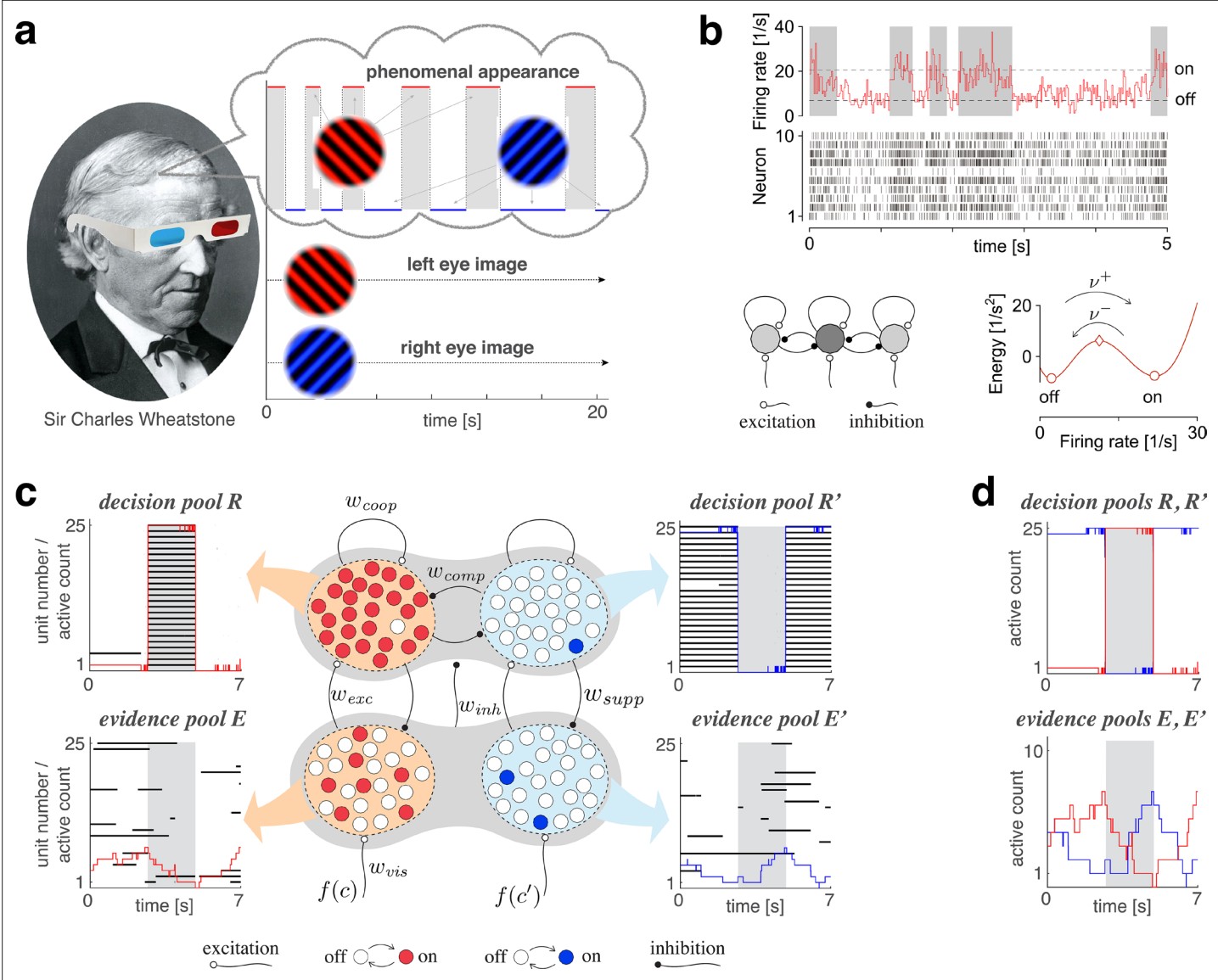

**Figure 1.** Proposed mechanism of binocular rivalry. (**a**) When the left and right eyes see incompatible images in the visual field, phenomenal appearance reverses at irregular intervals, sometimes being dominated by one image and sometimes by the other (gray and white regions). Sir Charles Wheatstone studied this multistable percept with a mirror stereoscope (not as shown!). (**b**) Spiking neural network implementation of a 'local attractor.' An assembly of 150 neurons (schematic, dark gray circle) interacts competitively with multiple other assemblies (light gray circles). Population activity of the assembly explores an effective energy landscape (right) with two distinct steady states (circles), separated by a ridge (diamond). Driven by noise, activity transitions occasionally between 'on' and 'off' states (bottom), with transition rates $\nu^{\pm}$ depending sensitively on external input to the assembly (not shown). Here, $\nu^{+} = \nu^{-} \approx 1\,Hz$. Spike raster shows 10 representative neurons. (**c**) Nested attractor dynamics (central schematic) that quantitatively reproduces the dynamics of binocular rivalry (left and right columns). Independently bistable variables ('local attractors,' small circles) respond probabilistically to input, transitioning stochastically between on- and off-states (red/blue and white, respectively). The entire system comprises four pools, with 25 variables each, linked by excitatory and inhibitory projections. Phenomenal appearance is decided by competition between decision pools $R$ and $R'$ forming 'non-local attractors' (cross-inhibition $w_{comp}$ and self-excitation $w_{coop}$). Visual input $c$ and $c'$ accumulates, respectively, in evidence pools $E$ and $E'$ and propagates to decision pools (feedforward selective excitation $w_{exc}$ and indiscriminate inhibition $w_{inh}$). Decision pools suppress associated evidence pools (feedback selective suppression $w_{supp}$). The time course of the number of active variables (active count) is shown for decision pools (top left and right) and evidence pools (bottom left and right), representing the left eye (red traces) and the right eye image (blue traces). The state of individual variables (black horizontal traces in left and middle columns) and of perceptual dominance (gray and white regions) is also shown. In decision pools, almost all variables become active (black trace) or inactive (no trace) simultaneously. In evidence pools, only a small fraction of variables is active at any given time. (**d**) Fractional activity dynamics of decision pools $R$ and $R'$ (top, red and blue traces) and evidence pools $E$ and $E'$ (bottom, red and blue traces). Reversals of phenomenal appearance are also indicated (gray and white regions).

### Relaxation dynamics

While activity $x(t)$ develops discretely and stochastically according to *Equation 5* (Materials and methods), its expectation $\langle x(t) \rangle$ develops continuously and deterministically,

$$\langle \dot{x} \rangle = \left(1 - \langle x \rangle\right) \nu^+ - \langle x \rangle \nu^- \tag{2}$$

relaxing with characteristic time $\tau_x = \frac{1}{\nu^+ + \nu^-}$ towards asymptotic value $x_\infty = \frac{\nu^+}{\nu^+ + \nu^-}$. As rates $\nu^\pm$ change with input $s$ (*Equation 1*), we can define the functions $\tau_s = \Upsilon(s)$ and $x_\infty = \Phi(s)$ (see Materials and methods). Characteristic time $\tau_x$ is longest for small input $s \simeq 0$ and shortens for larger positive or negative input $|s| \gg 0$. The asymptotic value $x_\infty$ ranges over the interval $(0, 1)$ and varies sigmoidally with input $s$, reaching half-activation for $s = -u^0/w$.

## Quality of representation

Pools of bistable variables belong to a class of neural representations particularly suited for Bayesian integration of sensory information (*Beck et al., 2008*; *Pouget et al., 2013*). In general, summation of activity is equivalent to optimal integration of information, provided that response variability is Poisson-like, and response tuning differs only multiplicatively (*Ma et al., 2006*; *Ma et al., 2008*). Pools of bistable variables closely approximate these properties (see Appendix 1, section Quality of representation: Suitability for inference).

The representational accuracy of even a comparatively small number of bistable variables can be surprisingly high. For example, if normally distributed inputs drive the activity of initially inactive pools of bistable variables, pools as used in the present model ($N = 25$, $w = 2.5$) readily capture 90% of the Fisher information (see Appendix 1, section Quality of representation: Integration of noisy samples).

## Conflicting evidence

Any model of BR must represent the conflicting evidence from both eyes (e.g., different visual orientations), which supports alternative perceptual hypotheses (e.g., distinct grating patterns). We assume that conflicting evidence accumulates in two separate pools of $N = 25$ bistable variables, $E$ and $E'$, ('evidence pools,' *Figure 1c*). Fractional activations $e(t)$ and $e'(t)$ develop stochastically following *Equation 5* (Materials and methods). Transition rates $\nu_e^\pm$ and $\nu_{e'}^\pm$ vary exponentially with activation energy (*Equation 1*), with baseline potential $u_e^0$ and baseline rate $\nu_e$. The variable components of activation energy, $u_e$ and $u_{e'}$, are synaptically modulated by image contrasts, $c$ and $c'$:

$$\begin{aligned} u_e &= w_{vis}\, I, \\ u_{e'} &= w_{vis}\, I' \end{aligned} \tag{3}$$

where $w_{vis}$ is a coupling constant and $I = f(c) \in [0, 1]$ is a monotonic function of image contrast $c$ (see Materials and methods).

## Competing hypotheses: 'non-local attractors'

Once evidence for, and against, alternative perceptual hypotheses (e.g., distinct grating patterns) has been accumulated, reaching a decision requires a sensitive and reliable mechanism for identifying the best supported hypothesis and amplifying the result into a categorical read-out. Such a winner-take-all decision (*Koch and Ullman, 1985*) is readily accomplished by a dynamical version of biased competition (*Deco and Rolls, 2005*; *Wang, 2002*; *Deco et al., 2007*; *Wang, 2008*).

We assume that alternative perceptual hypotheses are represented by two further pools of $N = 25$ bistable variables, $R$ and $R'$, forming two 'non-local attractors' ('decision pools,' *Figure 1c*). Similar to previous models of decision-making and attentional selection (*Deco and Rolls, 2005*; *Wang, 2002*; *Deco et al., 2007*; *Wang, 2008*), we postulate recurrent excitation within pools, but recurrent inhibition between pools, to obtain a 'winner-take-all' dynamics. Importantly, we assume that 'evidence pools' project to 'decision pools' not only in the form of selective excitation (targeted at the corresponding decision pool), but also in the form of indiscriminate inhibition (targeting both decision pools), as suggested previously (*Ditterich et al., 2003*; *Bogacz et al., 2006*).

Specifically, fractional activations $r(t)$ and $r'(t)$ develop stochastically according to *Equation 5* (Materials and methods). Transition rates $\nu_s^\pm$ and $\nu_{s'}^\pm$ vary exponentially with activation energy

(*Equation 1*), with baseline difference $u_r^0$ and baseline rate $\nu_r$. The variable components of activation energy, $u_r$ and $u_{r'}$, are synaptically modulated by evidence and decision activities:

$$
\begin{aligned}
u_r &= w_{exc}\, e - w_{inh}\left(e + e'\right) + w_{coop}\, r - w_{comp}\, r' \\
u_{r'} &= w_{exc}\, e' - w_{inh}\left(e + e'\right) + w_{coop}\, r' - w_{comp}\, r
\end{aligned}
\tag{4}
$$

where coupling constants $w_{exc}$, $w_{inh}$, $w_{coop}$, $w_{comp}$ reflect feedforward excitation, feedforward inhibition, lateral cooperation within decision pools, and lateral competition between decision pools, respectively.

This biased competition circuit expresses a categorical decision by either raising $r$ towards unity (and lowering $r'$ towards zero) or vice versa. The choice is random when visual input is ambiguous, $I \simeq I'$, but becomes deterministic with growing input bias $|I - I'| > 0$. This probabilistic sensitivity to input bias is reliable and robust under arbitrary initial conditions of $e$, $e'$, $r$ and $r'$ (see Appendix 1, section Categorical choice with *Appendix 1—figure 3*).

## Feedback suppression

Finally, we assume feedback suppression, with each decision pool selectively targeting the corresponding evidence pool. A functional motivation for this systematic bias *against* the currently dominant appearance is given momentarily. Its effects include curtailment of dominance durations and ensuring that reversals occur from time to time. Specifically, we modify *Equation 3* to

$$
\begin{aligned}
u_e &= w_{vis}\, f(c) - w_{supp}\, r \\
u_{e'} &= w_{vis}\, f(c') - w_{supp}\, r'
\end{aligned}
\tag{3a}
$$

where $w_{supp}$ is a coupling constant.

Previous models of BR (*Dayan, 1998*; *Hohwy et al., 2008*) have justified selective feedback suppression of the evidence supporting a winning hypothesis in terms of 'predictive coding' and 'hierarchical Bayesian inference' (*Rao and Ballard, 1999*; *Lee and Mumford, 2003*). An alternative normative justification is that, in volatile environments, where the sensory situation changes frequently ('volatility prior'), optimal inference requires an exponentially growing bias *against* evidence for the most likely hypothesis (*Veliz-Cuba et al., 2016*). Note that feedback suppression applies selectively to evidence for a winning hypothesis and is thus materially different from visual adaptation (*Wark et al., 2009*), which applies indiscriminately to all evidence present.

## Reversal dynamics

A representative example of the joint dynamics of evidence and decision pools is illustrated in *Figure 1c,d*, both at the level of pool activities $e(t)$, $e'(t)$, $r(t)$, $r'(t)$, and at the level of individual bistable variables $x(t)$. The top row shows decision pools $R$ and $R'$, with instantaneous active counts, $Nr(t)$ and $Nr'(t)$ and active/inactive states of individual variables $x(t)$. The bottom row shows evidence pools $E$ and $E'$, with instantaneous active counts, $Ne(t)$ and $Ne'(t)$ and active/inactive states of individual variables $x(t)$. Only a small fraction of evidence variables is active at any one time.

Phenomenal appearance reverses when the differential activity $\Delta e = e - e'$ of evidence pools, $E$ and $E'$, *contradicts* sufficiently strongly the differential activity $\Delta r = r - r'$ of decision pools, $R$ and $R'$, such that the steady state of decision pools is destabilized (see further below and Figure 4). As soon as the reversal has been effected at the decision level, feedback suppression *lifts from* the newly non-dominant evidence and *descends upon* the newly dominant evidence. Due to this asymmetric suppression, the newly non-dominant evidence recovers, whereas the newly dominant evidence habituates. This opponent dynamics progresses, past the point of equality $s \simeq s'$, until differential evidence activity $\Delta e$ once again *contradicts* differential decision activity $\Delta r$. Whereas the activity of decision pools varies *in phase* (or counterphase) with perceptual appearance, the activity of evidence pools changes *in quarterphase* (or negative quarterphase) with perceptual appearance (e.g., *Figures 1c,d, 2a*), consistent with some previous models (*Gigante et al., 2009*; *Albert et al., 2017*; *Weilnhammer et al., 2017*).

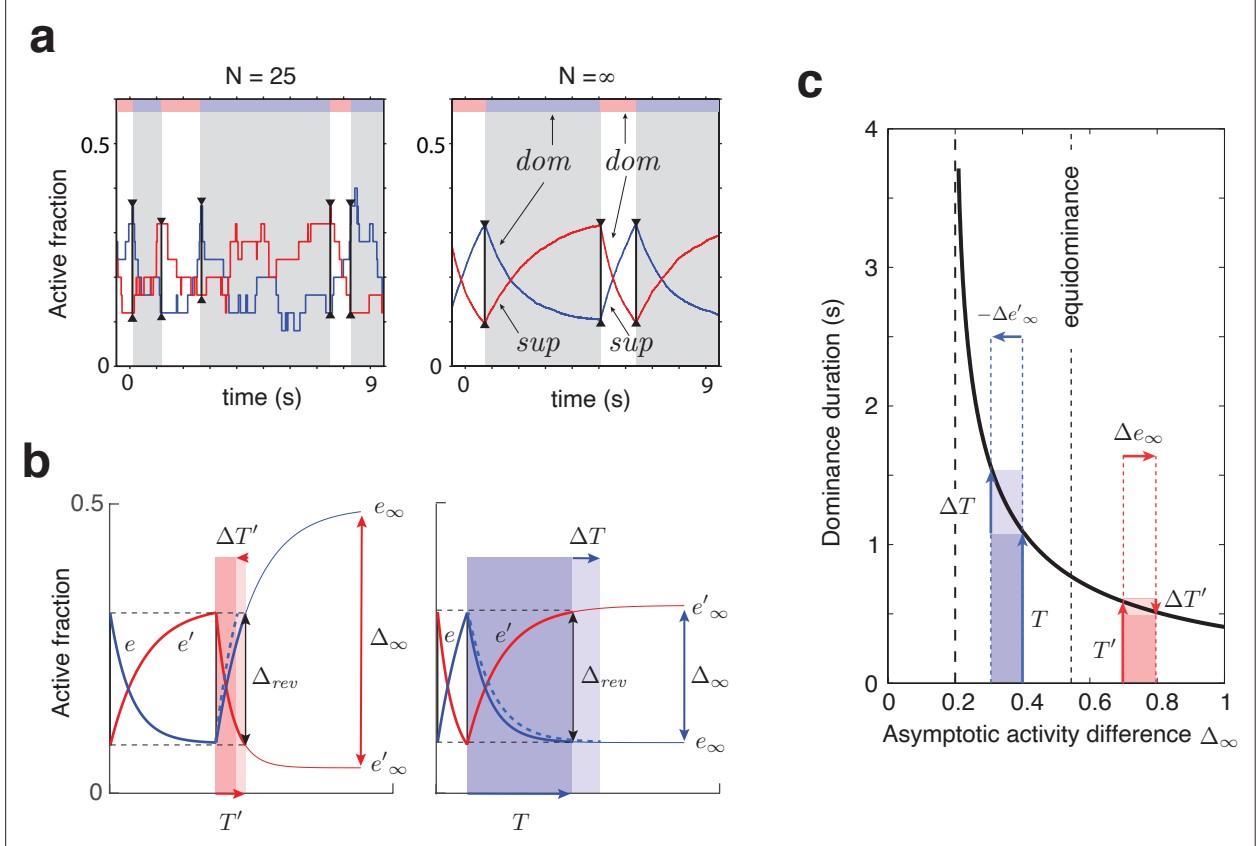

**Figure 2.** Joint dynamics of evidence habituation and recovery. Exponential development of evidence activities is governed by input-dependent asymptotic values and characteristic times. (**a**) Fractional activities $e$ (blue traces) and $e'$ (red traces) of evidence pools $E$ and $E'$, respectively, over several dominance periods for unequal stimulus contrast ($c = \frac{7}{8}, c' = \frac{1}{8}$). Stochastic reversals of finite system ($N = 25$ units per pool, left) and deterministic reversals of infinite system ($N$, right). Perceptual dominance (decision activity) is indicated along the upper margin (red or blue stripe). Dominance evidence habituates (*dom*), and non-dominant evidence recovers (*sup*), until evidence contradicts perception sufficiently (black vertical lines) to trigger a reversal (gray and white regions). (**b**) Development of stronger-input evidence $e$ (blue) and weaker-input evidence $e'$ (red) over two successive dominance periods ($c = \frac{15}{16}, c' = \frac{1}{16}$). Activities recover, or habituate, exponentially until reversal threshold $\Delta_{rev}$ is reached. Thin curves extrapolate to the respective asymptotic values, $e_\infty$ and $e'_\infty$. Dominance durations depend on distance $\Delta_\infty$ and on characteristic times $\tau_e$ and $\tau_{e'}$. Left: incrementing non-dominant evidence $e$ (dashed curve) raises upper asymptotic value $e_\infty$ and shortens dominance $T'$ by $\Delta T'$. Right: incrementing dominant evidence $e$ (dashed curve) raises lower asymptotic value $e_\infty$ and shortens dominance $T$ by $\Delta T$. (**c**) Increasing asymptotic activity difference $\Delta_\infty$ accelerates the development of differential activity and curtails dominance periods $T$, $T'$ (and vice versa). As the dependence is hyperbolic, any change to $\Delta_\infty$ disproportionately affects longer dominance periods. If $T > T'$, then $\Delta T > \Delta T'$ (and vice versa).

## Binocular rivalry

To compare predictions of the model described above to experimental observations, we measured spontaneous reversals of BR for different combinations of image contrast. BR is a particularly well-studied instance of multistable perception (*Wheatstone, 1838*; *Diaz-Caneja, 1928*; *Levelt, 1965*; *Leopold and Logothetis, 1999*; *Brascamp et al., 2015*). When conflicting images are presented to each eye (e.g., by means of a mirror stereoscope or of colored glasses, see Materials and methods), the phenomenal appearance reverses from time to time between the two images (*Figure 1a*). Importantly, the perceptual conflict involves also representations of coherent (binocular) patterns and is not restricted to eye-specific (monocular) representations (*Logothetis et al., 1996*; *Kovács et al., 1996*; *Bonneh et al., 2001*; *Blake and Logothetis, 2002*).

Specifically, our experimental observations established reversal sequences for $5 \times 5$ combinations of image contrast, $c_{dom}, c_{sup} \in \{\frac{1}{16}, \frac{1}{8}, \frac{1}{4}, \frac{1}{2}, 1\}$. During any given dominance period, $c_{dom}$ is the contrast of the phenomenally dominant image and $c_{sup}$ the contrast of the other, phenomenally suppressed image (see Materials and methods). We analyzed these observations in terms of mean dominance

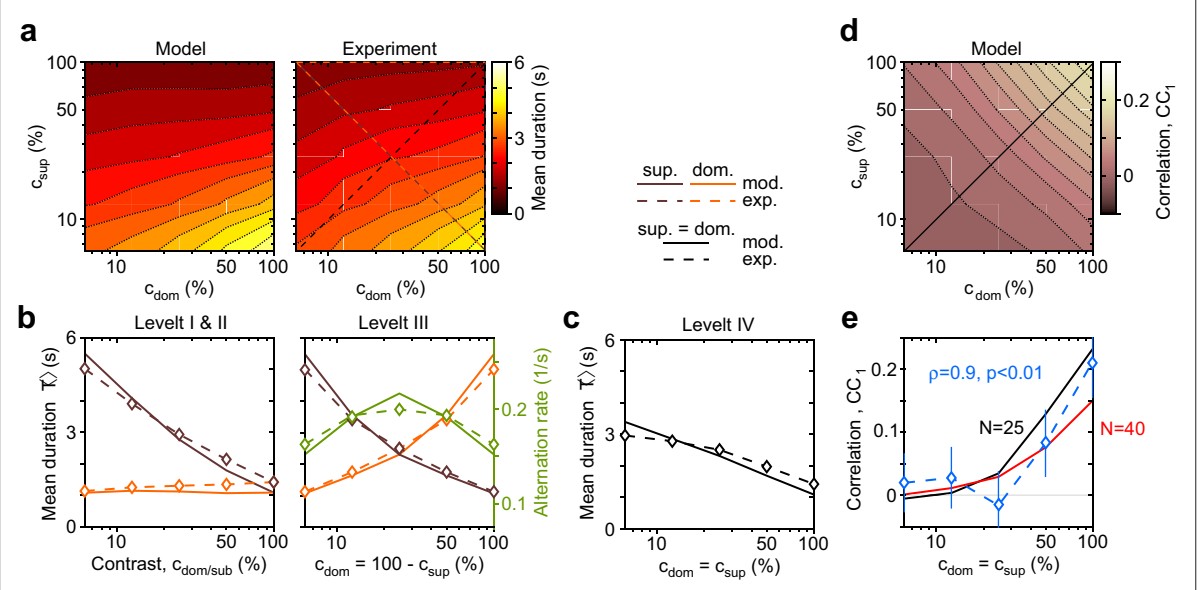

**Figure 3.** Dependence of mean dominance duration on dominant and suppressed image contrast ('Levelt's propositions'). (**a**) Mean dominance duration $\langle T \rangle$ (color scale), as a function of dominant contrast $c_{dom}$ and suppressed contrast $c_{sup}$, in model (left) and experiment (right). (**b**) Model prediction (solid traces) and experimental observation (dashed traces and symbols) compared. Levelt I and II: weak increase of $\langle T \rangle$ with $c_{dom}$ when $c_{sup} = 1$ (red traces and symbols), and strong decrease with $c_{sup}$ when $c_{dom} = 1$ (brown traces and symbols). Levelt III: symmetric increase of $\langle T \rangle$ with $c_{dom}$ (orange traces and symbols) and decrease with $c_{sup}$ (brown traces and symbols), when $c_{dom} + c_{dom} = 1$. Alternation rate (green traces and symbols) peaks at equidominance and decreases symmetrically to either side. (**c**) Levelt IV: decrease of $\langle T \rangle$ with image contrast, when $c_{sup} = c_{dom}$. (**d**) Predicted dependence of sequential correlation $cc_1$ (color scale) on $c_{dom}$ and $c_{sup}$. (**e**) Model prediction (black trace, $N = 25$) and experimental observation (blue trace and symbols, mean ± SEM, Spearman's rank correlation $\rho$), when $c_{sup} = c_{dom}$. Also shown is a second model prediction (red trace, $N = 40$).

durations $\langle T \rangle$, higher moments $c_V$ and $\gamma_1/c_V$ of the distribution of dominance durations, and sequential correlation $cc_1$ of successive dominance durations.

Additional aspects of serial dependence are discussed further below.

As described in Materials and methods, we fitted 11 model parameters to reproduce observations with more than 50 degrees of freedom: $5 \times 5$ mean dominance durations $\langle T \rangle$, $5 \times 5$ coefficients of variation $c_V$, one value of skewness $\gamma_1/c_V = 2$, and one correlation coefficient $cc_1 = 0.06$. The latter two values were obtained by averaging over $5 \times 5$ contrast combinations and rounding. Importantly, minimization of the fit error, by random sampling of parameter space with a stochastic gradient descent, resulted in a three-dimensional manifold of suboptimal solutions. This revealed a high degree of redundancy among the 11 model parameters (see Materials and methods). Accordingly, we estimate that the effective number of degrees of freedom needed to reproduce the desired out-of-equilibrium dynamics was between 3 and 4. Model predictions and experimental observations are juxtaposed in *Figures 3 and 4*.

The complex and asymmetric dependence of mean dominance durations on image contrast — aptly summarized by Levelt's 'propositions' I to IV (*Levelt, 1965*; *Brascamp et al., 2015*) — is fully reproduced by the model (*Figure 3*). Here, we use the updated definition of *Brascamp et al., 2015*: increasing the contrast of one image increases the fraction of time during which this image dominates appearance ('predominance,' Levelt I). Counterintuitively, this is due more to shortening dominance of the *unchanged* image than to lengthening dominance of the *changed* image (Levelt II, *Figure 3b*, left panel). Mean dominance durations grow (and alternation rates decline) symmetrically around equal predominance as contrast difference $c_{dom} - c_{sup}$ increases (Levelt III, *Figure 3b*, right panel). Mean dominance durations shorten when both image contrasts $c_{dom} = c_{sup}$ increase (Levelt IV, *Figure 3c*).

Successive dominance durations are typically correlated positively (*Fox and Herrmann, 1967*; *Walker, 1975*; *Pastukhov et al., 2013c*). Averaging over all contrast combinations, observed and fitted correlation coefficients were comparable with $cc_1 = 0.06 \pm 0.06$ (mean and standard deviation). Unexpectedly, both observed and fitted correlations coefficients increased systematically with image contrast ($\rho = 0.9$, $p < .01$), growing from $cc_1 = 0.02 \pm 0.05$ at $c_{dom} = c_{sup} = \frac{1}{16}$ to $0.21 \pm 0.06$ at $c_{dom} = c_{dom} = 1$ (*Figure 3e*, blue symbols). It is important to that this dependence was *not* fitted.

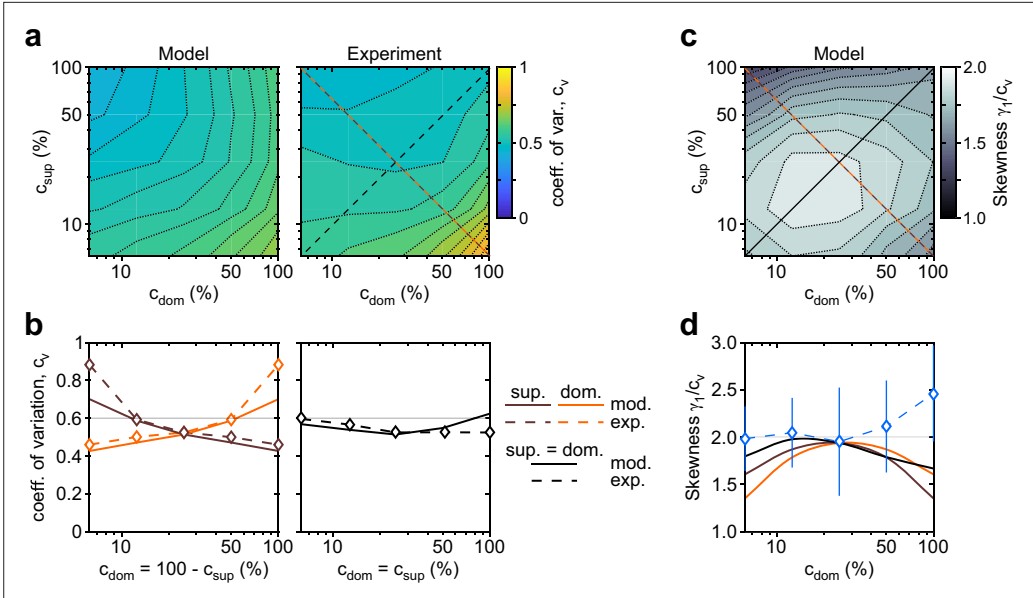

**Figure 4.** Shape of dominance distribution depends only weakly on image contrast ('scaling property'). Distribution shape is parametrized by coefficient of variation $c_v$ and relative skewness $\gamma_1/c_V$. (**a**) Coefficient of variation $c_v$ (color scale), as a function of dominant contrast $c_{dom}$ and suppressed contrast $c_{sup}$, in model (left) and experiment (right). (**b**) Model prediction (solid traces) and experimental observation (dashed traces and symbols) compared. Left: increase of $c_v$ with $c_{dom}$ (red traces and symbols), and symmetric decrease with $c_{sup}$ (brown traces and symbols), when $c_{sup} = 1$. Right: weak dependence when $c_{dom} = c_{sup}$ (black traces and symbols). (**c**) Predicted dependence of relative skewness $\gamma_1/c_V$ (gray scale) on $c_{dom}$ and $c_{sup}$. (**d**) Model prediction (solid traces), when $c_{dom} = c_{sup}$ (black) and $c_{dom} = 1 - c_{sup}$ (orange and brown) and experimental observation when $c_{dom} = c_{sup}$ (blue dashed trace and symbols, mean ± SEM).

Rather, this previously unreported dependence constitutes a model prediction that is confirmed by observation.

The distribution of dominance durations typically takes a characteristic shape (*Cao et al., 2016*; *Fox and Herrmann, 1967*; *Blake et al., 1971*; *Borsellino et al., 1972*; *Walker, 1975*; *De Marco et al., 1977*; *Murata et al., 2003*; *Brascamp et al., 2005*; *Pastukhov and Braun, 2007*; *Denham et al., 2018*), approximating a gamma distribution with shape parameter $r \simeq 3 - 4$, or coefficient of variation $c_V = 1/\sqrt{r} \simeq 0.5 - 0.6$. The fitted model fully reproduces this 'scaling property' (*Figure 4*). The observed coefficient of variation remained in the range $c_V \simeq 0.05 - 0.06$ for nearly all contrast combinations (*Figure 4b*). Unexpectedly, both observed and fitted values increased above, or decreased below, this range at extreme contrast combinations (*Figure 4b*, left panel). Along the main diagonal $c_{dom} = c_{sup}$ , where observed values had smaller error bars, both observed and fitted values of skewness were $\gamma_1/c_V \simeq 2$ and thus approximated a gamma distribution (*Figure 4d*, blue symbols).

## Specific contribution of evidence and decision levels

What are the reasons for the surprising success of the model in reproducing universal characteristics of multistable phenomena, including the counterintuitive input dependence ('Levelt's propositions'), the stereotypical distribution shape ('scaling property'), and the positive sequential correlation (as detailed in *Figures 3 and 4*)? Which level of model dynamics is responsible for reproducing different aspects of BR dynamics?

Below, we describe the specific contributions of different model components. Specifically, we show that the evidence level of the model reproduces 'Levelt's propositions I–III' and the 'scaling property,' whereas the decision level reproduces 'Levelt's proposition IV.' A non-trivial interaction between evidence and decision levels reproduces serial dependencies. Additionally, we show that this interaction predicts further aspects of serial dependencies – such as sensitivity to image contrast – that were not reported previously, but are confirmed by our experimental observations.

## Levelt's propositions I, II, and III

The characteristic input dependence of average dominance durations emerges in two steps (as in *Gigante et al., 2009*). First, inputs and feedback suppression shape the birth-death dynamics of evidence pools $E$ and $E'$ (by setting disparate transition rates $\nu^{\pm}$, following Equation 3' and *Equation 1*). Second, this sets in motion two opponent developments (habituation of dominant evidence activity and recovery of non-dominant evidence activity, both following *Equation 2*) that jointly determine dominance duration.

To elucidate this mechanism, it is helpful to consider the limit of large pools ($N \rightarrow \infty$) and its deterministic dynamics (*Figure 2*), which corresponds to the *average* stochastic dynamics. In this limit, periods of dominant evidence $E$ or $E'$ start and end at the same levels ($e_{start} = e'_{start}$ and $e_{end} = e'_{end}$), because reversal thresholds $\Delta_{rev}$ are the same for evidence difference $e - e'$ and $e' - e$ (see section Levelt IV below).

The rates at which evidence habituates or recovers depend, in the first instance, on asymptotic levels $e_{\infty}$ and $e'_{\infty}$ (*Equation 1 and 2*, *Figure 2b* and *Appendix 1—figure 4*). In general, dominance durations depend on distance $\Delta_{\infty}$ between asymptotic levels: the further apart these are, the faster the development and the shorter the duration. As feedback suppression *inverts* the sign of the opponent developments, dominant evidence decreases (habituates) while non-dominant evidence increases (recovers). Due to this inversion, $\Delta_{\infty}$ is roughly proportional to $e_{\infty}^{non-dom} - e_{\infty}^{dom} + w_{supp}$. It follows that the distance $\Delta_{\infty}$ is *smaller* and the reversal dynamics *slower* when dominant input is *stronger*, and vice versa. It further follows that incrementing one input (and raising the corresponding asymptotic level) speeds up recovery or slows down habituation, shortening or lengthening periods of non-dominance and dominance, respectively (Levelt I).

In the second instance, rates of habituation or recovery depend on characteristic times $\tau_e$ and $\tau_{e'}$ (*Equation 1 and 2*). When these rates are unequal, dominance durations depend more sensitively on the *slower* process. This is why dominance durations depend more sensitively on non-dominant input (Levelt II): recovery of non-dominant evidence is generally slower than habituation of dominant evidence, independently of which input is weaker or stronger. The reason is that the respective effects of characteristic times $\tau_e$ and $\tau_{e'}$ and asymptotic levels $e_{\infty}$ and $e'_{\infty}$ are synergistic for weaker-input evidence (in both directions), whereas they are antagonistic for stronger-input evidence (see Appendix 1, section Deterministic dynamics: Evidence pools and *Appendix 1—figure 4*).

In general, dominance durations depend hyperbolically on $\Delta_{\infty}$ (*Figure 2c* and *Equation 7* in Appendix 1). Dominance durations become infinite (and reversals cease) when $\Delta_{\infty}$ falls below the reversal threshold $\Delta_{rev}$. This hyperbolic dependence is also why alternation rate peaks at equidominance (Levelt III): increasing the difference between inputs always lengthens longer durations more than it shortens shorter durations, thus lowering alternation rate.

## Distribution of dominance durations

For all combinations of image contrast, the mechanism accurately predicts the experimentally observed distributions of dominance durations. This is owed to the stochastic activity of pools of bistable variables.

Firstly, dominance distributions retain nearly the same shape, even though average durations vary more than threefold with image contrast (see also *Appendix 1—figure 6a,b*). This 'scaling property' is due to the Poisson-like variability of birth-death processes (see Appendix 1, section Stochastic dynamics). Generally, when a stochastic accumulation approaches threshold, the rates of both accumulation and dispersion of activity affect the distribution of first-passage-times (*Cao et al., 2014*; *Cao et al., 2016*). In the special case of Poisson-like variability, the two rates vary proportionally and preserve distribution shape (see also *Appendix 1—figure 6c,d*).

Secondly, predicted distributions approximate gamma distributions with scale factor $r \simeq 3 - 4$. As shown previously (*Cao et al., 2014*; *Cao et al., 2016*), this is due to birth-death processes accumulating activity within a narrow range (i.e., evidence difference $\Delta e \leq 0.2$). In this low-threshold regime, the first-passage-times of birth-death processes are both highly variable and gamma distributed, consistent with experimental observations.

Thirdly, the predicted variability (coefficients of variation) of dominance periods varies along the $c + c' = 1$ axis, being larger for longer than for shorter dominance durations (*Figure 4a,b*). The reason is that stochastic development becomes noise-dominated. For longer durations, stronger-input

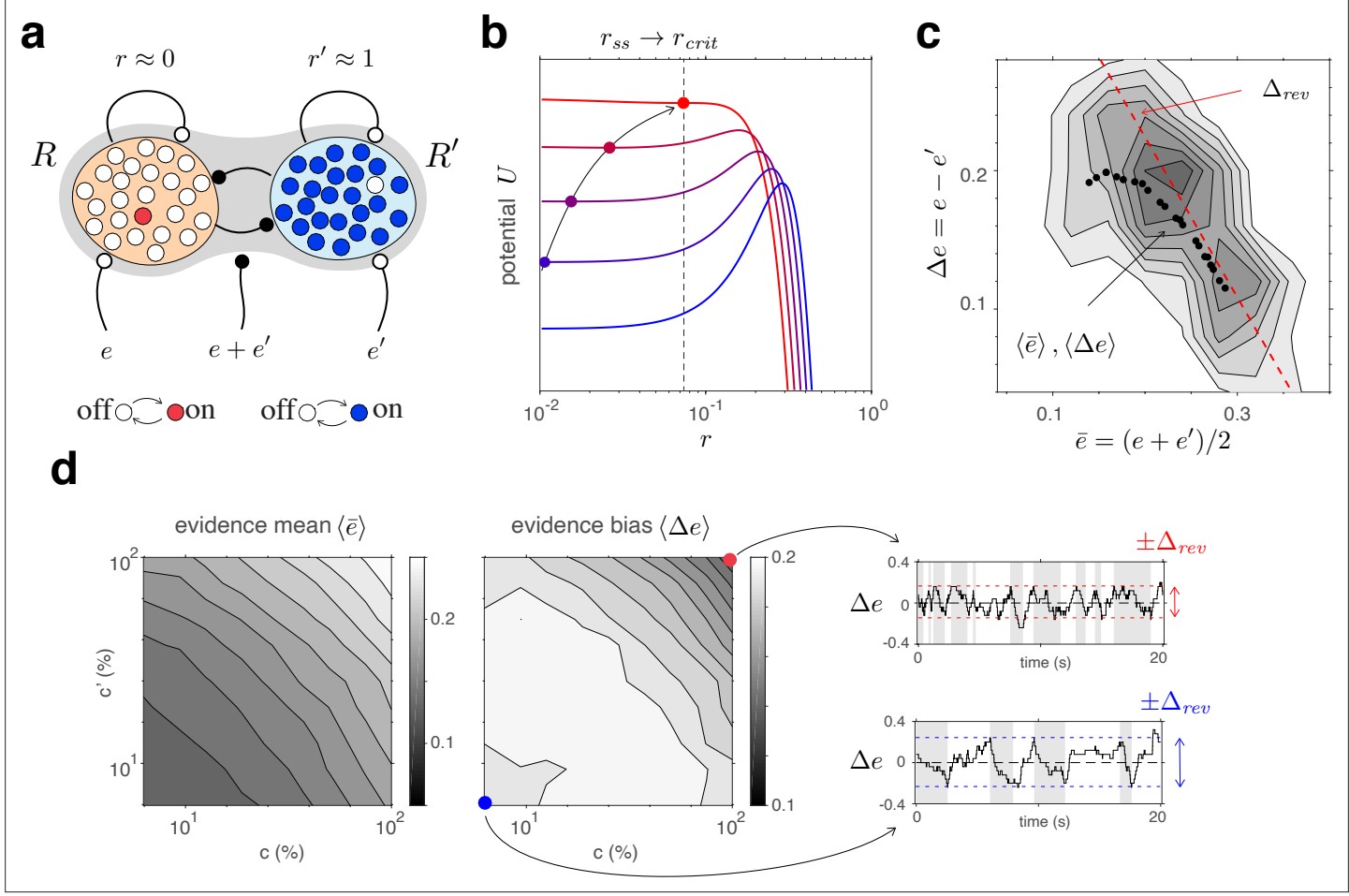

**Figure 5.** Competitive dynamics of decision pools ensures Levelt IV. (**a**) The joint stable state of decision pools (here $r' \simeq 1$ and $r \simeq 0$) can be destabilized by sufficiently contradictory evidence, $e > e'$. (**b**) Effective potential $U(e, e', r, r')$ (colored curves) and steady states $r_\infty$ (colored dots) for different levels of contradictory input, $\Delta e = e - e'$. Increasing $\Delta e$ destabilizes the steady state and shifts $r_\infty$ rightward (curved arrow). The critical value $r_{crit}$ (dotted vertical line), at which the steady state turns unstable, is reached when $\Delta e$ reaches the reversal threshold $\Delta_{rev}$. At this point, a reversal ensues with $r \to 1$ and $r' \to 0$. (**c**) The reversal threshold $\Delta_{rev}$ diminishes with combined evidence $e + e'$. In the deterministic limit, $\Delta_{rev}$ decreases linearly with $\bar{e} = (e + e')/2$ (dashed red line). In the stochastic system, the average evidence bias $\langle \Delta e \rangle$ at the time of reversals decreases similarly with the average evidence mean $\langle \bar{e} \rangle$ (black dots). Actual values of $\Delta e$ at the time of reversals are distributed around these average values (gray shading). (**d**) Average evidence mean $\langle \bar{e} \rangle$ (left) and average evidence bias $\langle \Delta e \rangle$ (middle) at the time of reversals as a function of image contrast $c$ and $c'$. Decrease of average evidence bias $\langle \Delta e \rangle$ with contrast shortens dominance durations (Levelt IV). At low contrast (blue dot), higher reversal thresholds $\Delta_{rev}$ result in less frequent reversals (bottom right, gray and white regions) whereas, at high contrast (red dot), lower reversal thresholds lead to more frequent reversals (top right).

evidence habituates rapidly into a regime where random fluctuations gain importance (see also *Appendix 1—figure 4a,b*).

## Levelt's proposition IV

The model accurately predicts how dominance durations shorten with higher image contrast $c = c'$ (Levelt IV). Surprisingly, this reflects the dynamics of decision pools $R$ and $R'$ (*Figure 5*).

Here again it is helpful to consider the deterministic limit of large pools ($N \to \infty$). In this limit, a dominant decision state $r' \simeq 1$ is destabilized when a *contradictory* evidence difference $\Delta e = e - e'$ exceeds a certain threshold value $\Delta_{rev}$ (*Figure 5b* and Appendix 1, section Deterministic dynamics: Decision pools). Due to the combined effect of excitatory and inhibitory feedforward projections, $w_{exc}$ and $w_{inh}$ (*Equation 4* and *Figure 5a*), this average reversal threshold *decreases* with mean evidence activity $\bar{e} = (e + e')/2$. Simulations of the fully stochastic model ($N = 25$) confirm this analysis

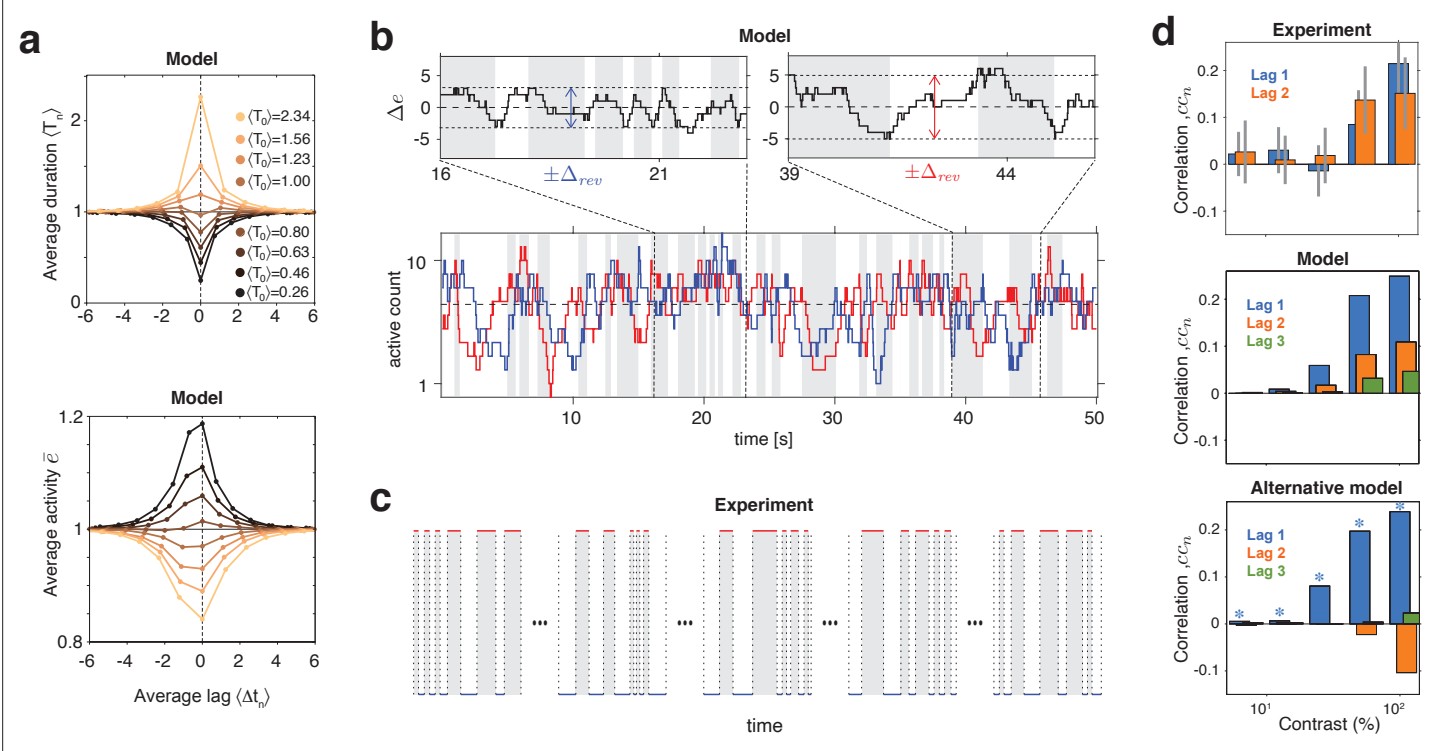

**Figure 6.** Serial dependency predicted by model and confirmed by experimental observations. (**a**) Conditional expectation of dominance duration $\langle T_{\pm n}\rangle$ (top) and of average mean evidence activity, $\langle \bar{e}_{\pm n}\rangle$ (bottom), in model simulations with maximal stimulus contrast ($c = c' = 1$). Dominance periods $T_0$ were grouped into octiles, from longest (yellow) to shortest (black). For each octile, the average duration $\langle T_{\pm n}\rangle$ of preceding and following dominance periods, as well as the average mean evidence activity $\langle \bar{e}_{\pm n}\rangle$ at the end of each period, is shown. All times in multiples of the overall average duration, $\langle T\rangle$, and activities in multiples of the overall average activity $\langle \bar{e}\rangle$. (**b**) Example reversal sequence from model. Bottom: stochastic development of evidence activities $e$ and $e'$ (red and blue traces), with large, joint fluctuations raising or lowering mean activity $\bar{e} = (e + e')/2$ above or below long-term average (dashed line). Top left: episode with $\bar{e}$ above average, *lower* $\Delta_{rev}$, and *shorter* dominance periods. Top right: episode with $\bar{e}$ below average, *higher* $\Delta_{rev}$, and *longer* dominance durations. (**c**) Examples of reversal sequences from human observers ($c = c' = 1$ and $c = c' = 1/2$). (**d**) Positive lagged correlations predicted by model (mean, middle) and confirmed by experimental observations (mean ± std, top). Alternative model (*Laing and Chow, 2002*) with adaptation and noise (mean, bottom), fitted to reproduce the values of $\langle T\rangle$, $c_v$, $\gamma_1$, and $cc_1$ predicted by the present model (blue stars).

(*Figure 5c*). As average evidence activity $\langle \bar{e}\rangle$ increases with image contrast, the average evidence bias $\langle \Delta e\rangle$ at the time of reversals decreases, resulting in shorter dominance periods (*Figure 5d*).

## Serial dependence

The proposed mechanism predicts positive correlations between successive dominance durations, a well-known characteristic of multistable phenomena (*Fox and Herrmann, 1967*; *Walker, 1975*; *Van Ee, 2005*; *Denham et al., 2018*). In addition, it predicts further aspects of serial dependence not reported previously.

In both model and experimental observations, a long dominance period tends to be followed by another long period, and a short dominance period by another short period (*Figure 6*). In the model, this is due to mean evidence activity $\bar{e} = (e + e')/2$ fluctuating stochastically above and below its long-term average. The autocorrelation time of these fluctuations increases monotonically with image contrast and, for high contrast, spans multiple dominance periods (see Appendix 1, section Characteristic times and *Appendix 1—figure 7*). Note that fluctuations of $\bar{e}$ diminish as the number of bistable variables increases and vanishe in the deterministic limit $N \to \infty$.

Crucially, fluctuations of mean evidence $\bar{e}$ modulate both reversal threshold $\Delta_{rev}$ and dominance durations $T$, as illustrated in *Figure 6a,b*. To obtain *Figure 6a*, dominance durations were grouped into quantiles and the average duration $\langle T_0\rangle$ of each quantile was compared to the conditional expectation of preceding and following durations $\langle T_{\pm n}\rangle$ (upper graph). For the same quantiles (compare color coding), average evidence activity $\langle \bar{e}_0\rangle$ was compared to the conditional expectation $\langle \bar{e}_{\pm n}\rangle$ at the

end of preceding and following periods (lower graph). Both the *inverse* relation between $\langle T_{\pm n} \rangle$ and $\langle \bar{e}_{\pm n} \rangle$ and the autocorrelation over multiple dominance periods are evident.

This source of serial dependency – comparatively slow fluctuations of $\bar{e}$ and $\Delta_{rev}$ – predicts several qualitative characteristics not reported previously and now confirmed by experimental observations. First, sequential correlations are predicted (and observed) to be strictly positive at all lags (next period, one-after-next period, and so on) (*Figure 6d*). In other words, it predicts that several successive dominance periods are shorter (or longer) than average.

Second, due to the contrast dependence of autocorrelation time, sequential correlations are predicted (and observed) to increase with image contrast (*Figure 6d*). The experimentally observed degree of contrast dependence is broadly consistent with pool sizes between $N = 25$ and $N = 40$ (black and red curves in *Figure 3e*). Larger pools with hundreds of bistable variables do not express the observed dependence on contrast (not shown).

Third, for high image contrast, reversal sequences are predicted (and observed) to contain extended episodes with dominance periods that are short or extended episodes with periods that are long (*Figure 6c*). When quantified in terms of a 'burstiness index,' the degree of inhomogeneity in predicted and observed reversal sequences is comparable (see Appendix 1, section Burstiness and *Appendix 1—figure 8*).

Many previous models of BR (e.g., *Laing and Chow, 2002*) postulated selective adaptation of competing representations to account for serial dependency. However, selective adaptation is an *opponent* process that favors *positive* correlations between *different* dominance periods, but *negative* correlations between *same* dominance periods. To demonstrate this point, we fitted such a model to reproduce our experimental observations ($T$, $c_V$, $\gamma_1$, and $cc_1$) for five image contrasts $c = c'$. As expected, the alternative model predicts *negative* correlations $cc_2$ for *same* dominance periods (*Figure 6d*, right panel), contrary to what is observed.

## Discussion

We have shown that many well-known features of BR are reproduced, and indeed guaranteed, by a particular dynamical mechanism. Specifically, this mechanism reproduces the counterintuitive input dependence of dominance durations ('Levelt's propositions'), the stereotypical shape of dominance distributions ('scaling property'), and the positive sequential correlation of dominance periods. The explanatory power of the proposed mechanism is considerably higher than that of previous models. Indeed, the observations explained exhibited more effective degrees of freedom (approximately 14) than the mechanism itself (between 3 and 4).

The proposed mechanism is biophysically plausible in terms of the out-of-equilibrium dynamics of a modular and hierarchical network of spiking neurons (see also further below). Individual modules idealize the input dependence of attractor transitions in assemblies of spiking neurons. All synaptic effects superimpose linearly, consistent with extended mean-field theory for neuronal networks (*Amit and Brunel, 1997*; *Van Vreeswijk and Sompolinski, 1996*). The interaction between 'rivaling' sets of modules ('pools') results in divisive normalization, which is consistent with many cortical models (*Carandini and Heeger, 2011*; *Miller, 2016*).

It has long been suspected that multistable phenomena in visual, auditory, and tactile perception may share a similar mechanistic origin. As the features of BR explained here are in fact universal features of multistable phenomena in different modalities, we hypothesize that similar out-of-equilibrium dynamics of modular networks may underlie all multistable phenomena in all sensory modalities. In other words, we hypothesize that this may be a general mechanism operating in many perceptual representations.

### Dynamical mechanism

Two principal alternatives have been considered for the dynamical mechanism of perceptual decision-making: drift-diffusion models (*Luce, 1986*; *Ratcliff and Smith, 2004*) and recurrent network models (*Wang, 2008*; *Wang, 2012*). The mechanism proposed here *combines* both alternatives: at its evidence level, sensory information is integrated, over both space and time, by 'local attractors' in a discrete version of a drift-diffusion process. At its decision level, the population dynamics of a recurrent network implements a winner-take-all competition between 'non-local attractors.' Together, the

two levels form a 'nested attractor' system (*Braun and Mattia, 2010*) operating perpetually out of equilibrium.

A recurrent network with strong competition typically 'normalizes' individual responses relative to the total response (*Miller, 2016*). Divisive normalization is considered a canonical cortical computation (*Carandini and Heeger, 2011*), for which multiple rationales can be found. Here, divisive normalization is augmented by indiscriminate feedforward inhibition. This combination ensures that decision activity rapidly and reliably categorizes *differential* input strength, largely independently of *total* input strength.

Another key feature of the proposed mechanism is that a 'dominant' decision pool applies feedback suppression to the associated evidence pool. Selective suppression of evidence for a winning hypothesis features in computational theories of 'hierarchical inference' (*Rao and Ballard, 1999*; *Lee and Mumford, 2003*; *Parr and Friston, 2017b*; *Pezzulo et al., 2018*), as well as in accounts of multistable perception inspired by such theories (*Dayan, 1998*; *Hohwy et al., 2008*; *Weilnhammer et al., 2017*). A normative reason for feedback suppression arises during continuous inference in uncertain and volatile environments, where the accumulation of sensory information is ongoing and cannot be restricted to appropriate intervals (*Veliz-Cuba et al., 2016*). Here, optimal change detection requires an exponentially rising bias *against* evidence for the most likely state, ensuring that even weak changes are detected, albeit with some delay.

The pivotal feature of the proposed mechanism are pools of bistable variables or 'local attractors.' Encoding sensory inputs in terms of persistent 'activations' of local attractors assemblies (rather than in terms of transient neuronal spikes) creates an intrinsically retentive representation: sites that respond are also sites that retain information (for a limited time). Our results are consistent with a few tens of bistable variables in each pool. In the proposed mechanism, *differential* activity of two pools accumulates evidence *against* the dominant appearance until a threshold is reached and a reversal ensues (see also *Barniv and Nelken, 2015*; *Nguyen et al., 2020*). Conceivably, this discrete nonequilibrium dynamics might instantiate a variational principle of inference such as 'maximum caliber' (*Pressé et al., 2013*; *Dixit et al., 2018*).

## Emergent features

The components of the proposed mechanism interact to guarantee the statistical features that characterize BR and other multistable phenomena. Discretely stochastic accumulation of differential evidence *against* the dominant appearance ensures sensitivity of dominance durations to non-dominant input. It also ensures the invariance of relative variability ('scaling property') and gamma-like distribution shape of dominance durations. Due to a non-trivial interaction with the competitive decision, discretely stochastic fluctuations of evidence-level activity express themselves in a serial dependency of dominance durations. Several features of this dependency were unexpected and not reported previously, for example, the sensitivity to image contrast and the 'burstiness' of dominance reversals (i.e., extended episodes in which dominance periods are consistently longer or shorter than average). The fact that these predictions are confirmed by our experimental observations provides further support for the proposed mechanism.

## Relation to previous models

How does the proposed mechanism compare to previous 'dynamical' models of multistable phenomena? It is of similar complexity as previous minimal models (*Laing and Chow, 2002*; *Wilson, 2007*; *Moreno-Bote et al., 2010*) in that it assumes four state variables at two dynamical levels, one slow (accumulation) and one fast (winner-take-all competition). It differs in reversing their ordering: visual input impinges first on the slow level, which then drives the fast level. It also differs in that stochasticity dominates the slow dynamics (as suggested by *van Ee, 2009*), not the fast dynamics. However, the most fundamental difference is discreteness (pools of bistable variables), which shapes all key dynamical properties.

Unlike many previous models (e.g., *Laing and Chow, 2002*; *Wilson, 2007*; *Moreno-Bote et al., 2007*; *Moreno-Bote et al., 2010*; *Cohen et al., 2019*), the proposed mechanism does not include adaptation (stimulation-driven weakening of evidence), but a phenomenologically similar feedback suppression (perception-driven weakening of evidence). Evidence from perceptual aftereffects supports the existence of both stimulation- and perception-driven adaptation, albeit at different levels

of representation. Aftereffects in the perception of simple visual features – such as orientation, spatial frequency, or direction of motion (*Blake and Fox, 1974*; *Lehmkuhle and Fox, 1975*; *Wade and Wenderoth, 1978*) – are driven by stimulation rather than by perceived dominance, whereas aftereffects in complex features – such as spiral motion, subjective contours, rotation in depth (*Wiesenfelder and Blake, 1990*; *Van der Zwan and Wenderoth, 1994*; *Pastukhov et al., 2014a*) – typically depend on perceived dominance. Several experimental observations related to BR have been attributed to stimulation-driven adaptation (e.g., negative priming, flash suppression, generalized flash suppression; *Tsuchiya et al., 2006*). The extent to which a perception-driven adaptation could also explain these observations remains an open question for future work.

Multistable perception induces a positive priming or 'sensory memory' (*Pearson and Clifford, 2005*; *Pastukhov and Braun, 2008*; *Pastukhov et al., 2013a*), which can stabilize a dominant appearance during intermittent presentation (*Leopold et al., 2003*; *Maier et al., 2003*; *Sandberg et al., 2014*). This positive priming exhibits rather different characteristics (e.g., shape-, size- and motion-specificity, inducement period, persistence period) than the negative priming/adaptation of rivaling representations (*de Jong et al., 2012*; *Pastukhov et al., 2013a*; *Pastukhov and Braun, 2013b*; *Pastukhov et al., 2014a*; *Pastukhov et al., 2014b*; *Pastukhov, 2016*). To our mind, this evidence suggest that sensory memory is mediated by additional levels of representation and not by self-stabilization of rivaling representations, as has been suggested (*Noest et al., 2007*; *Leptourgos, 2020*). To incorporate sensory memory, the present model would have to be extended to include three hierarchical levels (evidence, decision, and memory), as previously proposed by *Gigante et al., 2009*.

BR arises within local regions of the visual field, measuring approximately $0.25°$ to $0.5°$ in the fovea (*Leopold, 1997*; *Logothetis, 1998*). No rivalry ensues when the stimulated locations in the left and right eye are more distant from each other. The computational model presented here encompasses only one such local region, and therefore cannot reproduce spatially extended phenomena such as piecemeal rivalry (*Blake et al., 1992*) or traveling waves (*Wilson et al., 2001*). To account for these phenomena, the visual field would have to be tiled with replicant models linked by grouping interactions (*Knapen et al., 2007*; *Bressloff and Webber, 2012*).

A particularly intriguing previous model (*Wilson, 2003*) postulated a hierarchy with competing and adapting representations in eight state variables at two separate levels, one lower (monocular) and another higher (binocular) level. This 'stacked' architecture could explain the fascinating experimental observation that one image can continue to dominate (dominance durations $\sim 2\,s$) even when images are rapidly swapped between eyes (period $1/3\,s$) (*Kovács et al., 1996*; *Logothetis et al., 1996*). We expect that our hierarchical model could also account for this phenomenon if it were to be replicated at two successive levels. It is tempting to speculate that such 'stacking' might have a normative justification in that it might subserve hierarchical inference (*Yuille and Kersten, 2006*; *Hohwy et al., 2008*; *Friston, 2010*).

Another previous model (*Li et al., 2017*) used a hierarchy with 24 state variables at three separate levels to show that a stabilizing influence of selective visual attention could also explain slow rivalry when images are swapped rapidly. Additionally, this rather complex model reproduced the main features of Levelt's propositions, but did not consider scaling property and sequential dependency. The model shared some of the key features of the present model (divisive inhibition, differential excitation-inhibition), but added a multiplicative attentional modulation. As the present model already incorporates the 'biased competition' that is widely thought to underlie selective attention (*Sabine and Ungerleider, 2000*; *Reynolds and Heeger, 2009*), we expect that it could reproduce attentional effects by means of additive modulations.

## Continuous inference

The notion that multistable phenomena such as BR reflect active exploration of explanatory hypotheses for sensory evidence has a venerable history (*von Helmholtz, 1867*; *Barlow et al., 1972*; *Gregory, 1980*; *Leopold and Logothetis, 1999*). The mechanism proposed here is in keeping with that notion: higher-level 'explanations' compete for control ('dominance') of phenomenal appearance in terms of their correspondence to lower-level 'evidence.' An 'explanation' takes control if its correspondence is sufficiently superior to that of rival 'explanations.' The greater the superiority, the longer control is retained. Eventually, alternative 'explanations' seize control, if only briefly. This manner of

operation is also consistent with computational theories of 'analysis by synthesis' or 'hierarchical inference,' although there are many differences in detail (*Rao and Ballard, 1999*; *Parr and Friston, 2017b*; *Pezzulo et al., 2018*).

Interacting with an uncertain and volatile world necessitates continuous and concurrent evaluation of sensory evidence and selection of motor action (*Cisek and Kalaska, 2010*; *Gold and Stocker, 2017*). Multistable phenomena exemplify continuous decision-making without external prompting (*Braun and Mattia, 2010*). Sensory decision-making has been studied extensively, mostly in episodic choice-task, and the neural circuits and activity dynamics underlying episodic decision-making – including representations of potential choices, sensory evidence, and behavioral goals – have been traced in detail (*Cisek and Kalaska, 2010*; *Gold and Shadlen, 2007*; *Wang, 2012*; *Krug, 2020*). Interestingly, there seems to be substantial overlap between choice representations in decision-making and in multistable situations (*Braun and Mattia, 2010*).

Continuous inference has been studied extensively in auditory streaming paradigms (*Winkler et al., 2012*; *Denham et al., 2014*). The auditory system seems to continually update expectations for sound patterns on the basis of recent experience. Compatible patterns are grouped together in auditory awareness, and incompatible patterns result in spontaneous reversals between alternatives. Many aspects of this rich phenomenology are reproduced by computational models driven by some kind of 'prediction error' (*Mill et al., 2013*). The dynamics of two recent auditory models (*Barniv and Nelken, 2015*; *Nguyen et al., 2020*) are rather similar to the model presented here: while one sound pattern dominates awareness, evidence *against* this pattern is accumulated at a subliminal level.

## Relation to neural substrate

What might be the neural basis of the bistable variables/'local attractors' proposed here? Ongoing activity in sensory cortex appears to be low-dimensional, in the sense that the activity of neurons with similar response properties varies concomitantly ('shared variability,' 'noise correlations,' *Ponce-Alvarez et al., 2012*, *Mazzucato et al., 2015*, *Engel et al., 2016*, *Rich and Wallis, 2016*, *Mazzucato et al., 2019*). This shared variability reflects the spatial clustering of intracortical connectivity (*Muir and Douglas, 2011*; *Okun et al., 2015*; *Cossell et al., 2015*; *Lee et al., 2016*; *Rosenbaum et al., 2017*) and unfolds over moderately slow time scales (in the range of $100\ ms$ to $500\ ms$) both in primates and rodents (*Ponce-Alvarez et al., 2012*; *Mazzucato et al., 2015*; *Cui et al., 2016*; *Engel et al., 2016*; *Rich and Wallis, 2016*; *Mazzucato et al., 2019*).

Possible dynamical origins of shared and moderately slow variability have been studied extensively in theory and simulation (for reviews, see *Miller, 2016*; *Huang and Doiron, 2017*; *La Camera et al., 2019*). Networks with weakly clustered connectivity (e.g., 3% rewiring) can express a metastable attractor dynamics with moderately long time scales (*Litwin-Kumar and Doiron, 2012*; *Doiron and Litwin-Kumar, 2014*; *Schaub et al., 2015*; *Rosenbaum et al., 2017*). In a metastable dynamics, individual (connectivity-defined) clusters transition spontaneously between distinct and quasi-stationary activity levels ('attractor states') (*Tsuda, 2001*; *Stern et al., 2014*).

Evidence for metastable attractor dynamics in cortical activity is accumulating steadily (*Mattia et al., 2013*; *Mazzucato et al., 2015*; *Rich and Wallis, 2016*; *Engel et al., 2016*; *Marcos et al., 2019*; *Mazzucato et al., 2019*). Distinct activity states with exponentially distributed durations have been reported in sensory cortex (*Mazzucato et al., 2015*; *Engel et al., 2016*), consistent with noise-driven escape transitions (*Doiron and Litwin-Kumar, 2014*; *Huang and Doiron, 2017*). And several reports are consistent with external input modulating cortical activity mostly indirectly, via the rate of state transitions (*Fiser et al., 2004*; *Churchland et al., 2010*; *Mazzucato et al., 2015*; *Engel et al., 2016*; *Mazzucato et al., 2019*).

The proposed mechanism assumes bistable variables with noise-driven escape transitions, with transition rates modulated exponentially by external synaptic drive. Following previous work (*Cao et al., 2016*), we show this to be an accurate reduction of the population dynamics of metastable networks of spiking neurons.

Unfortunately, the spatial structure of the 'shared variability' or 'noise correlations' in cortical activity described above is poorly understood. However, we estimate that the cortical representation of our rivaling display involves approximately $400\ mm^2$ and $200\ mm^2$ of cortical surface in cortical areas V1 and V4, respectively (*Winawer and Witthoft, 2015*; *Winawer and Benson, 2021*). Accordingly, in each of these two cortical areas, the neural representation of rivaling stimulation can comfortably

accommodate several thousand recurrent local assemblies, each capable of expressing independent collective dynamics (i.e., 'classic columns' comprising several 'minicolumns' with distinct stimulus selectivity *Nieuwenhuys R, 1994*, *Kaas, 2012*). Thus, our model assumes that the representation of two rivaling images engages approximately 1–2% of the available number of recurrent local assemblies.

## Neurophysiological correlates of BR

Neurophysiological correlates of BR have been studied extensively, often by comparing reversals of phenomenal appearance during binocular stimulation with physical alternation (PA) of monocular stimulation (e.g., *Leopold and Logothetis, 1996*; *Scheinberg and Logothetis, 1997*; *Logothetis, 1998*; *Wilke et al., 2006*; *Aura et al., 2008*; *Keliris et al., 2010*; *Panagiotaropoulos et al., 2012*; *Bahmani et al., 2014*; *Xu et al., 2016*; *Kapoor et al., 2020*; *Dwarakanath et al., 2020*). At higher cortical levels, such as inferior temporal cortex (*Scheinberg and Logothetis, 1997*) or prefrontal cortex (*Panagiotaropoulos et al., 2012*; *Kapoor et al., 2020*; *Dwarakanath et al., 2020*), BR and PA elicit broadly comparable neurophysiological responses that mirror perceptual appearance. Specifically, activity crosses its average level at the time of each reversal, roughly *in phase* with perceptual appearance (*Scheinberg and Logothetis, 1997*; *Kapoor et al., 2020*). In primary visual cortex (area V1), where many neurons are dominated by input from one eye, neurophysiological correlates of BR and PA diverge in an interesting way: whereas modulation of spiking activity is weaker during BR than PA (*Leopold and Logothetis, 1996*; *Logothetis, 1998*; *Wilke et al., 2006*; *Aura et al., 2008*; *Keliris et al., 2010*), measures thought to record dendritic inputs are modulated comparably under both conditions (*Aura et al., 2008*; *Keliris et al., 2010*; *Bahmani et al., 2014*; *Yang et al., 2015*; *Xu et al., 2016*). A stronger divergence is observed at an intermediate cortical level (visual area V4), where neurons respond to both eyes. Whereas some units modulate their spiking activity comparably during BR and PA (i.e., *increased* activity when preferred stimulus becomes dominant), other units exhibit the opposite modulation during BR (i.e., *reduced* activity when preferred stimulus gains dominance) (*Leopold and Logothetis, 1996*; *Logothetis, 1998*; *Wilke et al., 2006*). Importantly, at this intermediate cortical level, activity crosses its average level well before and after each reversal (*Leopold and Logothetis, 1996*; *Logothetis, 1998*), roughly *in quarter phase* with perceptual appearance.

Some of these neurophysiological observations are directly interpretable in terms of the model proposed here. Specifically, activity modulation at higher cortical levels (inferotemporal cortex, prefrontal cortex) could correspond to 'decision activity,' predicted to vary *in phase* with perceptual appearance. Similarly, activity modulation at intermediate cortical levels (area V4) could correspond to 'evidence activity,' which is predicted to vary *in quarter phase* with perceptual appearance. This identification would also be consistent with the neurophysiological evidence for attractor dynamics in columns of area V4 (*Engel et al., 2016*). The subpopulation of area V4 with opposite modulation could mediate feedback suppression from decision levels. If so, our model would predict this subpopulation to vary *in counterphase* with perceptual appearance. Finally, the fascinating interactions observed within primary visual cortex (area V1) are well beyond the scope of our simple model. Presumably, a 'stacked' model with two successive levels of competitive interactions at monocular and binocular levels or representation (*Wilson, 2003*; *Li et al., 2017*) would be required to account for these phenomena.

## Conclusion

As multistable phenomena and their characteristics are ubiquitous in visual, auditory, and tactile perception, the mechanism we propose may form a general part of sensory processing. It bridges neural, perceptual, and normative levels of description and potentially offers a 'comprehensive task-performing model' (*Kriegeskorte and Douglas, 2018*) for sensory decision-making.

# Materials and methods
## Psychophysics

Six practiced observers participated in the experiment (four males, two females). Informed consent, and consent to publish, was obtained from all observers, and ethical approval Z22/16 was obtained from the Ethics Commission of the Faculty of Medicine of the Otto-von-Guericke University, Magdeburg. Stimuli were displayed on an LCD screen (EIZO ColorEdge CG303W, resolution $2560 \times 1600$

pixels, viewing distance was 104 cm, single pixel subtended $0.014°$, refresh rate 60 Hz) and were viewed through a mirror stereoscope, with viewing position being stabilized by chin and head rests. Display luminance was gamma-corrected and average luminance was $50\ cd/m^2$.

Two grayscale circular orthogonally oriented gratings ($+45°$ and $-45°$) were presented foveally to each eye. Gratings had diameter of $1.6°$, spatial period $2\ cyc/deg$. To avoid a sharp outer edge, grating contrast was modulated with Gaussian envelope (inner radius $0.6°$, $\sigma = 0.2°$). Tilt and phase of gratings was randomized for each block. Five contrast levels were used: 6.25, 12.5, 25, 50, and 100%. Contrast of each grating was systematically manipulated, so that each contrast pair was presented in two blocks (50 blocks in total). Blocks were $120\ s$ long and separated by a compulsory 1 min break. Observers reported on the tilt of the visible grating by continuously pressing one of two arrow keys. They were instructed to press only during exclusive visibility of one of the gratings, so that mixed percepts were indicated by neither key being pressed (25% of total presentation time). To facilitate binocular fusion, gratings were surrounded by a dichoptically presented square frame (outer size 9.8°, inner size 2.8°).

Dominance periods of 'clear visibility' were extracted in sequence from the final $90\ s$ of each block and the mean linear trend was subtracted from all values. Values from the initial $30\ s$ were discarded. To make comparable the dominance periods of different observers, values were rescaled by the ratio of the all-condition-all-observer average ($2.5\ s$) and the all-condition average of each observer ($2.5 \pm 1.3\ s$). Finally, dominance periods from symmetric conditions ($c_{left}, c_{right}$) with $c_{left} = c_{right}$ were combined into a single category ($c_{dom}, c_{sup}$), where $c_{dom}$ ($c_{sup}$) was the contrast viewed by the dominant (suppressed) eye. The number of observed dominance periods ranged from 900 to 1700 per contrast combination ($1300 \pm 240$).

For the dominance periods $T$ observed in each condition, first, second, and third central moments were computed, as well as coefficient of variation $c_V$ and skewness $\gamma_1$ relative to coefficient of variation:

$$\mu_1 = \langle T \rangle, \quad \mu_2 = \langle T^2 \rangle - \langle T \rangle, \quad \mu_3 = \langle T^3 \rangle - 3\langle T \rangle \langle T^2 \rangle + 2\langle T \rangle^3$$

$$c_V = \frac{\sqrt{\mu_2}}{\mu_1}, \qquad \frac{\gamma_1}{c_V} = \frac{\mu_3\,\mu_1}{\mu_2^2}$$

The expected standard error of the mean for distribution moments is 2% for the mean, 3% for the coefficient of variation, and 12% for skewness relative to coefficient of variation, assuming 1000 gamma-distributed samples.

Coefficients of sequential correlations were computed from pairs of periods $(T_i, T_j)$ with opposite dominance (first and next: 'lag' $j - i = 1$), pairs of periods with same dominance (first and next but one: 'lag' $j - i = 2$), and so on,

$$cc_k = \frac{\langle T_i - \langle T_i \rangle \rangle \, \langle T_j - \langle T_j \rangle \rangle}{\sqrt{\left(\langle T_i^2 \rangle - \langle T_i \rangle^2\right)\left(\langle T_j^2 \rangle - \langle T_j \rangle^2\right)}}$$

where $\langle T \rangle$ and $\langle T^2 \rangle$ are mean duration and mean square duration, respectively. The expected standard deviation of the coefficient of correlation is 0.03, assuming 1000 gamma-distributed samples.

To analyze 'burstiness,' we adapted a statistical measure used in neurophysiology (*Compte et al., 2003*). First, sequences of dominance periods were divided into all possible subsets of $k \in \{2, 3, \ldots, 16\}$ successive periods and mean durations computed for each subset. Second, heterogeneity was assessed by computing, for each size $k$, the coefficient of variation $c_v$ over mean durations, compared to the mean and variance of the corresponding coefficient of variation for randomly shuffled sequences of dominance periods. Specifically, a 'burstiness index' was defined for each subset size $k$ as.

$$BI(k) = \frac{c_V - \langle c_V \rangle_{shuffle}}{\sqrt{\langle c_V^2 \rangle_{shuffle} - \langle c_V \rangle_{shuffle}^2}}$$

where $c_V$ is the coefficient of variation over subsets of size $k$ and where $\langle c_V \rangle_{shuffle}$ and $\langle c_V^2 \rangle_{shuffle}$ are, respectively, mean and mean square of the coefficients of variation from shuffled sequences.

## Model

The proposed mechanism for BR dynamics relies on discretely stochastic processes ('birth-death' or generalized Ehrenfest processes). Bistable variables $x \in \{0,1\}$ transition between active and inactive states with time-varying Poisson rates $\nu^+(t)$ (activation) and $\nu^-(t)$ (inactivation). Two 'evidence pools' of $N$ such variables, $E$ and $E'$, represent two kinds visual evidence (e.g., for two visual orientations), whereas two 'decision pools,' $R$ and $R'$, represent alternative perceptual hypotheses (e.g., two grating patterns) (see also **Appendix 1—figure 1**). Thus, instantaneous dynamical state is represented by four active counts $n_e, n_{e'}, n_r, n_{r'} \in [0, N]$ or, equivalently, by four active fractions $e, e', r, r' \in [0, 1]$.

The development of pool activity over time is described by a master equation for probability $P_n(t)$ of the number $n(t) \in [0, N]$ active variables.

$$
\begin{aligned}
\partial_t P_n(t) &= (N - n + 1)\nu^+ P_{n-1}(t) + (n + 1)\nu^- P_{n+1}(t) \\
&\quad - [(N - n)\nu^+ + n\nu^-] P_n(t)
\end{aligned}
\tag{5}
$$

For constant $\nu^\pm$, the distribution $P_n(t)$ is binomial at all times **Karlin and McGregor, 1965**, **van Kampen, 1981**. The time development of the number of active units $n_X(t)$ in pool $X$ is an inhomogeneous Ehrenfest process and corresponds to the count of activations, minus the count of deactivations,

$$
\Delta n_X(t) = \underbrace{\mathcal{B}\left(N - n_X, \nu^+ \Delta t\right)}_{activations} - \underbrace{\mathcal{B}\left(n_X, \nu^- \Delta t\right)}_{inactivations}
$$

where $\mathcal{B}\left(n, \nu \Delta t\right)$ is a discrete random variable drawn from a binomial distribution with trial number $n$ and success probability $\nu \Delta t$.

All variables of a pool have identical transition rates, which depend exponentially on the 'potential difference' $\Delta u = u + u^0$ between states, with a input-dependent component $u$ and a baseline component $u^0$:

$$
\nu_s^\pm = \frac{\nu_s}{2} e^{\pm(u_e + u_e^0)/2}, \qquad \nu_{s'}^\pm = \frac{\nu_s}{2} e^{\pm(u_{e'} + u_e^0)/2}
$$

$$
\nu_r^\pm = \frac{\nu_r}{2} e^{\pm(u_r + u_r^0)/2}, \qquad \nu_{r'}^\pm = \frac{\nu_r}{2} e^{\pm(u_{r'} + u_r^0)/2}
$$

where $\nu_e$ and $\nu_r$ are baseline rates and $u_e^0$ and $u_r^0$ baseline components. The input-dependent components of effective potentials are modulated linearly by synaptic couplings

$$
\begin{aligned}
u_s &= w_{vis} f(c) - w_{supp}\, r \\
u_{s'} &= w_{vis} f(c') - w_{supp}\, r' \\
u_r &= w_{exc}\, e - w_{inh}\left(e + e'\right) + w_{coop}\, r - w_{comp}\, r' \\
u_{r'} &= w_{exc}\, e - w_{inh}\left(e + e'\right) + w_{coop}\, r' - w_{comp}\, r
\end{aligned}
$$

Visual inputs are $I = f(c)$ and $I' = f(c')$, respectively, where

$$
f(c) = \frac{\ln(1 + c/\gamma)}{\ln(1 + 1/\gamma)} \in \{0, 1\}
$$

is a monotonically increasing, logarithmic function of image contrast, with parameter $\gamma$.

## Degrees of freedom

The proposed mechanism has 11 independent parameters – 6 synaptic couplings, 2 baseline rates, 2 baseline potentials, 1 contrast nonlinearity – which were fitted to experimental observations. A 12th parameter – pool size – remained fixed.

| Symbol | Description | Value |
| --- | --- | --- |
| N | Pool size | 25 |

*Continued on next page*

*Continued*

| Symbol | Description | Value |
|---|---|---|
| $1/\nu_e$ | Baseline rate, evidence | 1.95 ± 0.10 s |
| $1/\nu_r$ | Baseline rate, decision | 0.018 ± 0.010 s |
| $u_e^0$ | Baseline potential, evidence | -1.65 ± 0.24 |
| $u_r^0$ | Baseline potential, decision | -4.94 ± 0.67 |
| $w_{vis}$ | Visual input coupling | 1.780 ± 0.092 |
| $w_{exc}$ | Feedforward excitation | 152.2 ± 3.7 |
| $w_{inh}$ | Feedforward inhibition | 32.10 ± 2.3 |
| $w_{comp}$ | Lateral competition | 33.4 ± 1.2 |
| $w_{coop}$ | Lateral cooperation | 15.21 ± 0.59 |
| $w_{supp}$ | Feedback suppression | 2.34 ± 0.14 |
| $\gamma$ | Contrast nonlinearity | 0.071 ± 0.011 |

## Fitting procedure

The experimental dataset consisted of two 5 × 5 arrays $X_i^{exp}$ for mean $\langle T \rangle$ and coefficient of variation $c_V$, plus two scalar values for skewness $\gamma_1 = 2$ and correlation coefficient $cc_1 = 0.06$. The two scalar values corresponded to the (rounded) average values observed over the 5 × 5 combinations of image contrast. In other words, the fitting procedure prescribed contrast dependencies for the first two distribution moments, but not for correlation coefficients.

The fit error $E_{fit}$ was computed as a weighted sum of relative errors

$$E_{fit} = \sum_{i=1}^{4} w_i \delta_i \bigg/ \sum_{i=1}^{4} w_i , \qquad \delta_i = \left| \frac{X_i^{mod} - X_i^{exp}}{\bar{X}_i^{exp}} \right|$$

with weighting $w = [1, 1, 1, \tfrac{1}{4}]$ emphasizing distribution moments.

Approximately 400 minimization runs were performed, starting from random initial configurations of model parameters. For the optimal parameter set, the resulting fit error for the *mean observer* dataset was approximately 13%. More specifically, the fit errors for mean dominance $\langle T \rangle$, coefficient of variation $c_V$, relative skewness $\gamma_1/c_V$, and correlation coefficients $cc_1$ and $cc_2$ were 9.8, 7.9, 8.7, 70, and 46%, respectively. Here, fit errors for relative skewness and correlation coefficients were computed for the isocontrast conditions, where experimental observations were least noisy.

To confirm that resulting fit was indeed optimal and could not be further improved, we studied the behavior of the fit error in the vicinity of the optimal parameter set. For each parameter $\alpha_i$, 30 values $\alpha_i^{(j)}$ were picked in the direct vicinity of the optimal parameter $\alpha_i^{opt}$ (**Appendix 1—figure 9**). The resulting scatter plot of value pairs $\alpha_i^{(j)}$ and fit error $E_{fit}^{(j)}$ was approximated by a quadratic function, which provided 95% confidence intervals for $\alpha_i^{(j)}$. For all parameters except $\nu_r$, the estimated quadratic function was convex and the coefficient of the Hessian matrix associated with the fit error was positive. Additionally, the estimated extremum of each parabola was close to the corresponding optimal parameter, confirming that the parameter set was indeed optimal (**Appendix 1—figure 9**).

To minimize fit error, we repeated a stochastic gradient descent from randomly chosen initial parameter. Interestingly, the ensemble of suboptimal solutions found by this procedure populated a low-dimensional manifold of the parameter space in three principal components accounted for 95% of the positional variance. Thus, models that reproduce experimental observations with varying degrees of freedom exhibit only 3–4 effective degrees of freedom. We surmise that this is due, on the one hand, to the severe constraints imposed by our model architecture (e.g., discrete elements, exponential input dependence of transition rates) and, on the other hand, by the requirement that the dynamical operating regime behaves as a relaxation oscillator.

In support of this interpretation, we note that our 5 × 5 experimental measurements of $\langle T \rangle$ and $c_V$ were accurately described by 'quadric surfaces' ($z = a_1 + a_2 x + a_3 y + a_4 x^2 + a_5 xy + a_6 y^2$) with six

coefficients each. Together with the two further measurements of $\gamma_1/c_V$ and $cc_1$, our experimental observations accordingly exhibited approximately $6 \times 2 + 2 = 14$ effective degrees of freedom. This number was sufficient to constrain the 3–4 dimensional manifold of parameters, where the model operated as a relaxation oscillator with a particular dynamics, specifically, a slow-fast dynamics associated, respectively, with the accumulation and reversal phases of BR.

## Alternative model

As an alternative model (**Laing and Chow, 2002**), a combination of competition, adaptation, and image-contrast-dependent noise was fitted to reproduce four $5 \times 5$ arrays $X_i^{exp}$ for mean $\langle T \rangle$, coefficient of variation $c_V$, skewness $\gamma_1$, and correlation coefficient $cc_1$. Fit error $E_{fit}$ was computed as the average of relative errors

$$E_{fit} = \frac{1}{n} \sum_{i=1}^{n} \delta_i, \qquad \delta_i = \left| \frac{X_i^{mod} - X_i^{exp}}{\bar{X}_i^{exp}} \right|$$

For purposes of comparison, a weighted fit error with weighting $w = [1, 1, 1, \frac{1}{4}]$ was computed, as well.

The model comprised four state variables and independent colored noise:

$$\tau_r \, \dot{r}_{1,2} = -r_{1,2} + F\left(-\beta r_{2,1} - \phi_a a_{1,2} + I_{1,2} + n_{1,2}\right)$$
$$\tau_a \, \dot{a}_{1,2} = -a_{1,2} + r_{1,2}$$
$$\tau_n \, \dot{n}_{1,2} = -n_{1,2} + \sigma_{1,2} \, \sqrt{2\tau_n} \, \xi(t)$$

where $F(x) = [1 + \exp(-x/\kappa)]^{-1}$ is a nonlinear activation function and $\xi(t)$ is white noise.

Additionally, both input $I_{1,2}$ and noise amplitude $\sigma_{1,2}$ were assumed to depend nonlinearly on image contrast $c_{1,2}$:

$$I_{1,2} = f(c_{1,2}) = b_I \, c_{1,2}^{k_I}, \qquad \sigma_{1,2} = g(c_{1,2}) = b_\sigma \, c_{1,2}^{k_\sigma}$$

This coupling between input and noise amplitude served stabilizes the shape of dominance distributions over different image contrasts ('scaling property').

Parameters for competition $\beta = 10$, activity time constant $\tau_r = 50 \, ms$, noise time constant $\tau_n = 500 \, ms$, and activation function $k = 0.1$ were fixed. Parameters for adaptation strength $\phi_a \in [1, 100]$, adaptation time constant $\tau_a \in [1, 00]$, contrast dependence of input $b_I \in [1, 5]$, $k_I \in [0.1, 5]$, and contrast dependence of noise amplitude $b_\sigma \in [0.1, 1]$, $k_\sigma \in [0.1, 1]$ were explored within the ranges indicated.

The best fit (determined with a genetic algorithm) was as follows: $\phi_a = 18.39$, $\tau_a = 22.78$, $k_I = 1.52$, $b_I = 2.92$, $k_\sigma = 0.57$, $b_\sigma = 0.19$. The fit errors for mean dominance $\langle T \rangle$, coefficient of variation $c_V$, skewness $\gamma_1$, and correlation coefficient $cc_1$ were, respectively, 11.3, 8.3, 20, and 55%. The fit error for correlation coefficient $cc_2$ was 180% (because the model predicted negative values). The combined average for $\langle T \rangle$, $c_V$, and $\gamma_1$ was 13.2%. The fit error obtained with weighting $w = (1, 1, 1, \frac{1}{4})$ was 16.4%.

For **Figure 6d**, the alternative model was fitted only to observations at equal image contrast, $c = c'$: mean dominance $\langle T \rangle$, coefficient of variation $c_V$, skewness $\gamma_1$, and correlation coefficient $cc_1$. The combined average fit error for $\langle T \rangle$, $c_V$, and $\gamma_1$ was 11.2%. The combined average for all four observables was 22%.

## Spiking network simulation

To illustrate a possible neural realization of 'local attractors,' we simulated a competitive network with eight identical assemblies of excitatory and inhibitory neurons, which collectively expresses a spontaneous and metastable dynamics (**Mattia et al., 2013**). One assembly (denoted as 'foreground') comprised 150 excitatory leaky-integrate-and-fire neurons, which were weakly coupled to the 1050 excitatory neurons of the other assemblies (denoted as 'background'), as well as 300 inhibitory neurons. Note that background assemblies are not strictly necessary and are included only for the sake of verisimilitude. The connection probability between any two neurons was $c = 2/3$. Excitatory synaptic efficacy between neurons in the same assembly and in two different assemblies was $J_{intra} = 0.612 mV$ and $J_{inter} = 0.403 mV$, respectively. Inhibitory synaptic efficacy was $J_I = -1.50 mV$, and the efficacy of

excitatory synapses onto inhibitory neurons was $J_{IE} = 0.560 mV$. Finally, 'foreground' neurons, 'background neurons,' and 'inhibitory neurons' each received independent Poisson spike trains of $2400 Hz$, $2280 Hz$ and $2400 Hz$, respectively. Other settings were as in *Mattia et al., 2013*. As a result of these settings, 'foreground' activity transitioned spontaneously between an 'off' state of approximately $4 Hz$ and an 'on' state of approximately $40 Hz$.

## Acknowledgements

Funding from EU FP7-269459 Coronet, DFG BR 987/3-1, DFG 987/4-1, and

EU Human Brain Project SGA3-945539.

The authors thank Andrew Parker and Maike S Braun for helpful comments.

## Additional information

### Funding

| Funder | Grant reference number | Author |
| --- | --- | --- |
| European Commission | FP7-269459 | Jochen Braun |
| Deutsche Forschungsgemeinschaft | BR 987/3-1 | Jochen Braun |
| Deutsche Forschungsgemeinschaft | BR 987/4-1 | Jochen Braun |
| H2020 European Research Council | 45539 | Maurizio Mattia |

The funders had no role in study design, data collection and interpretation, or the decision to submit the work for publication.

### Author contributions

Robin Cao, Conceptualization, Formal analysis, Funding acquisition, Investigation, Methodology, Software, Supervision, Visualization, Writing – original draft, Writing – review and editing; Alexander Pastukhov, Maurizio Mattia, Conceptualization, Data curation, Formal analysis, Investigation, Methodology, Software, Supervision, Visualization; Stepan Aleshin, Formal analysis, Investigation, Methodology, Software; Jochen Braun, Conceptualization, Funding acquisition, Methodology, Software, Supervision, Visualization, Writing – original draft, Writing – review and editing

### Author ORCIDs

Alexander Pastukhov ⓘ http://orcid.org/0000-0002-8738-8591
Maurizio Mattia ⓘ http://orcid.org/0000-0002-2356-4509
Jochen Braun ⓘ http://orcid.org/0000-0002-8886-078X

### Ethics

Human subjects: Six practised observers participated in the experiment (4 male, 2 female). Informed consent, and consent to publish, was obtained from all observers and ethical approval Z22/16 was obtained from the Ethics Commisson of the Faculty of Medicine of the Otto-von-Guericke University, Magdeburg.

### Decision letter and Author response

### Data availability

Source data is provided for Figures 2 and 3. Source code for the binocular rivalry model is provided in a Github repository (https://github.com/mauriziomattia/2021.BistablePerceptionModel) copy archived at https://archive.softwareheritage.org/swh:1:rev:f70e9e45ddb64cef7fc9a3ea57f0b7a04dfc6729.

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

## Appendix 1

### Model schematics

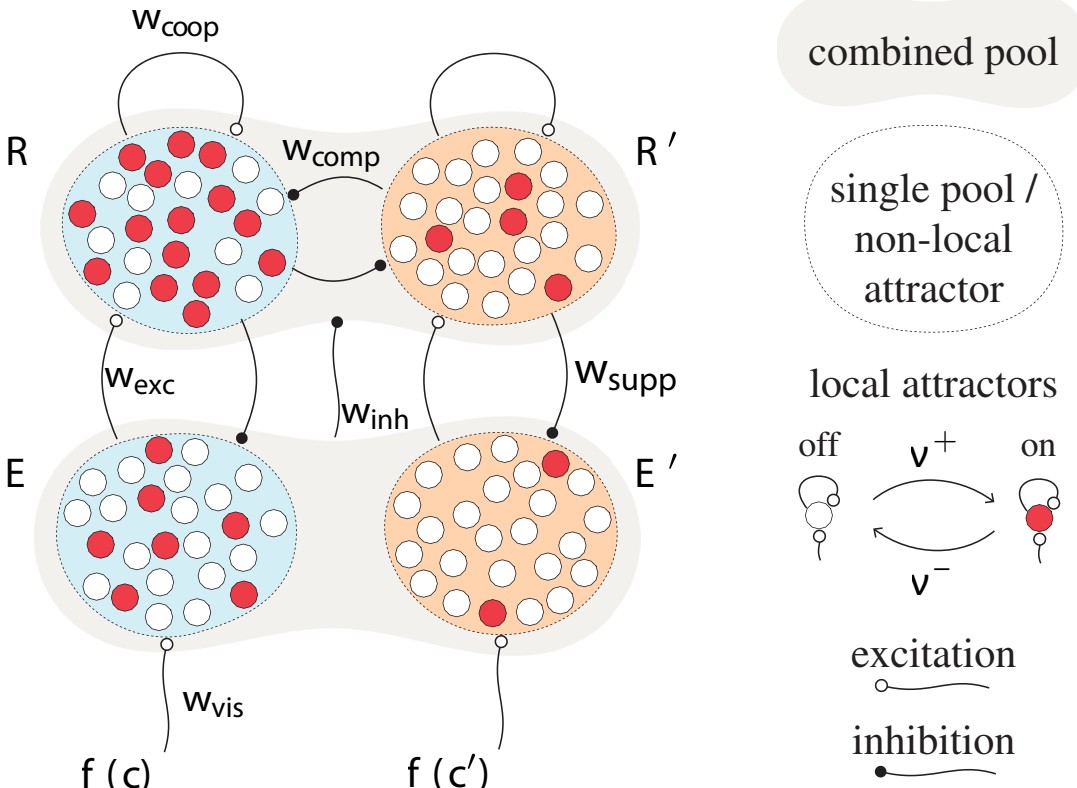

**Appendix 1—figure 1.** Proposed mechanism of binocular rivalry dynamics (schematic). Bistable variables are represented by white (inactive) or red (active) circles. Four pools, each with $N = 25$ variables, are shown: two evidence pools $E$ and $E'$, with active counts $n_e(t)$ and $n_{e'}(t)$, and two decision pools, $R$ and $R'$, with active counts $n_r(t)$ and $n_{r'}(t)$. Excitatory and inhibitory synaptic couplings include selective feedforward excitation $w_{exc}$, indiscriminate feedforward inhibition $w_{inh}$, recurrent excitation $w_{coop}$, and mutual inhibition $w_{comp}$ of decision pools, as well as selective feedback suppression $w_{supp}$ of evidence pools. Visual input to evidence pools $f(c)$ and $f(c')$ is a function of image contrast $c$ and $c'$.

Metastable attractor dynamics

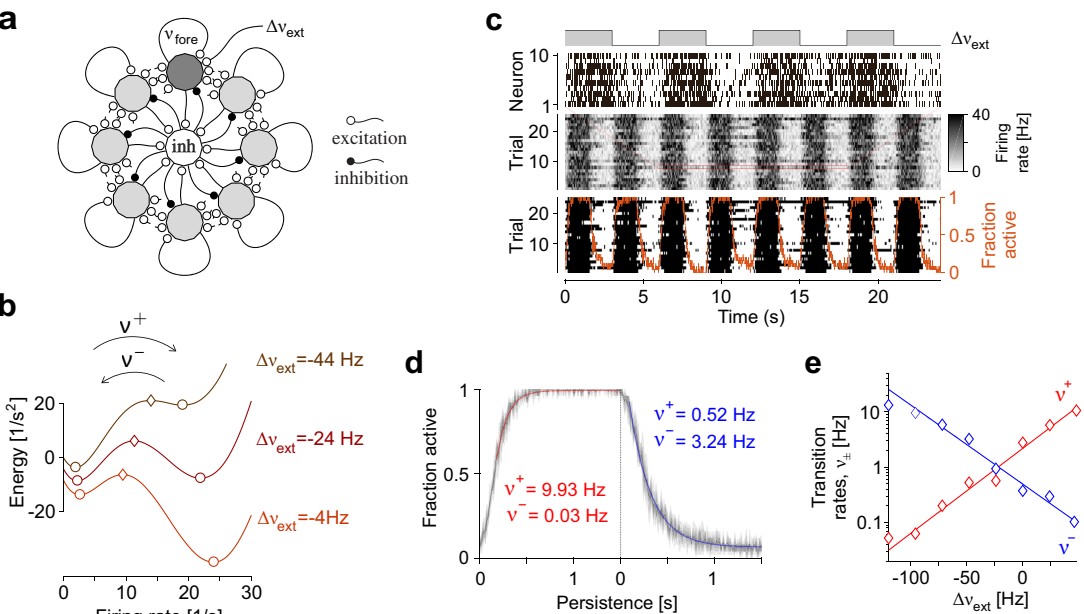

**Appendix 1—figure 2.** Metastable dynamics of spiking neural network. (**a**) Eight assemblies of excitatory neurons (schematic, light and dark gray disks) and one pool of inhibitory neurons (white disc) interact competitively with recurrent random connectivity. We focus on one 'foreground' assembly (dark gray), with firing rate $\nu_{fore}$ and selective external input $\Delta\nu_{ext}$. (**b**) 'Foreground' activity explores an effective energy landscape with two distinct steady states (circles), separated by ridge points (diamonds). As this landscape changes with external input $\Delta\nu_{ext}$, transition rates $\nu^{\pm}$ between 'on' and 'off' states also change with external input. (**c**) Simulation to establish transition rates $\nu^{\pm}$ of foreground assembly. External input $\Delta\nu_{ext}$ is stepped periodically between $-44\,Hz$ and $-4\,Hz$.

Spiking activity of 10 representative excitatory neurons in a single trial, population activity over 25 trials, thresholded population activity over 25 trials, and activation probability (fraction of 'on' states). (**d**) Relaxation dynamics in response to step change of $\Delta\nu_{ext}$, with 'on' transitions (left) and 'off' transitions (right). (**e**) Average state transition rates $\nu^{\pm}$ vary anti-symmetrically and exponentially with external input: $\nu^{+} \simeq 2.2\,Hz\exp\left(+0.8\,5s\,\Delta\nu_{ext}\right)$ and $\nu^{-} \simeq 0.5\,Hz\exp\left(-0.79\,s\,\Delta\nu_{ext}\right)$ (red and blue lines).

We postulate assemblies or clusters of neurons with recurrent random connectivity as operative units of sensory representations. In our model, such assemblies are reduced to binary variables with Poisson transitions. Our key assumption is that the rates $\nu^{\pm}$ of activation and inactivation events are modulated exponentially by synaptic input (**Equation 1**):

$$\nu^{\pm} = \nu\,e^{\pm(ws+u_0)}$$

Here, we show that these assumptions are a plausible reduction of recurrently connected assemblies of spiking neurons.

Following earlier work, we simulated a competitive network with eight identical assemblies of excitatory and inhibitory neurons (**Appendix 1—figure 2a**), configured to collectively express a metastable activity dynamics (**Mattia et al., 2013**). Here, we are interested particularly in the activity dynamics of one excitatory assembly (dubbed 'foreground'), which expresses two quasi-stable 'attractor' states: an 'on' state with high activity. In the context of the metastable network, the 'foreground' assembly is bistable in that it transitions spontaneously between 'on' and 'off' states. Such state transitions are noise-driven escape events from an energy well and therefore occur with Poisson-like rates $\nu^{+}$ (activation) and $\nu^{-}$ (inactivation). **Figure 1b** and **Appendix 1—figure 2b** illustrate this energy landscape for the 'diffusion limit' of very large assemblies, where quasi-stable activity levels are $\nu_{fore} \simeq 45\,Hz$ for the 'on' state and $\nu_{fore} \simeq 4\,Hz$ for the 'off' state. For small assemblies with fewer neurons, the difference between 'on' and 'off' states is less pronounced.

To establish the dependence of transition rates on external input to the 'foreground' assembly, we stepped external input rate $\Delta\nu_{ext}$ between two values selected from a range $\Delta\nu_{ext} \in [-120\,Hz, 50\,Hz]$ and monitored the resulting spiking activity in individual neurons, as well as activity $\nu_{fore}$ of the entire population (*Appendix 1—figure 2c*, upper and middle panels). Comparing population activity to a suitable threshold, we identified 'on' and 'off' states of the 'foreground' assembly (*Appendix 1—figure 2c*, lower panel), as well as the probability of 'on' or 'off' states at different points in time following a step in $\Delta\nu_{ext}$ (*Appendix 1—figure 2d*). From the hazard rate (temporal derivative of probability), we then estimated the rates $\nu^{\pm}$ of state transitions shown in *Appendix 1—figure 2d*. Transition rates $\nu^{\pm}$ vary approximately anti-symmetrically and exponentially with external input $\Delta\nu_{ext}$. In the present example, $\nu^+ \simeq 2.2\,Hz\exp\left(+0.85\,s\,\Delta\nu_{ext}\right)$ and $\nu^- \simeq 0.5\,Hz\exp\left(-0.79\,s\,\Delta\nu_{ext}\right)$ (*Appendix 1—figure 2e*, red and blue lines). This Arrhenius–Van't-Hoff-like dependence of escape rates is a consequence of the approximately linear dependence of activation energy on external input. Escape kinetics is typical for attractor systems and motivates *Equation 1*.

## Quality of representation

### Accumulation of information

A birth-death process – defined as $N$ bistable variables with transition rates $\nu^{\pm} = ve^{\pm ws}$, where $\nu$ is a baseline rate and $w$ a coupling constant – accumulates and retains information about input $s$, performing as a 'leaky integrator' with a characteristic time scale [*Braun and Mattia, 2010*]. Specifically, the value of $s$ may be inferred from fractional activity $x(t)$ at time $t$, if coupling $w$ and baseline rate $\nu$ are known. The inverse variance of the maximum likelihood estimate is given by the Fisher information

$$J_x(s,t) = \frac{N\left[\partial_s\langle x\rangle\right]^2}{\langle x\rangle(1 - \langle x\rangle)} \qquad (1)$$

Its value grows with time, approaching $J_x = Nw^2/cosh^2(ws/2)$ for $t \to \infty$. For small inputs $s \simeq 0$, the Fisher information increases monotonically as $J_x(t) \approx \left(Nw^2/4\right)\tanh\left(\nu t/2\right)$. Surprisingly, the upper bound of $J_x \le Nw^2/4$ depends linearly on pool size $N$, but quadratically on coupling $w$. Thus, stronger coupling substantially improves encoding accuracy (of input $s$).

The rate at which Fisher information is accumulated by a pool is set by the baseline transition rate $\nu$. An initially inactive pool, with $n_0 = 0$, accumulates Fisher information at an initial rate of $\partial_t J_x|_{t=0} = \left(\nu Nw^2/4\right)e^{ws/2}$. Thus, any desired rate of gaining Fisher information may be obtained by choosing an appropriate value for $\nu$. However, unavoidably, after an input $s$ has ceased (and was replaced by another), information about $s$ is lost at the same rate.

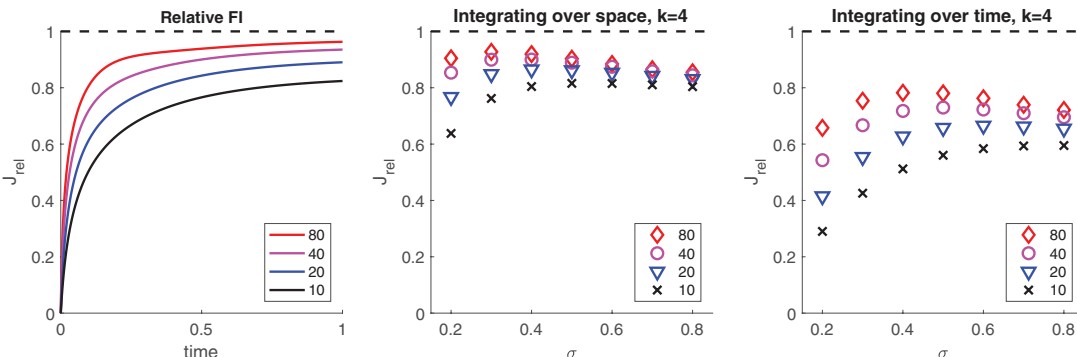

**Appendix 1—figure 3.** Information retained by stochastic pool activity from normally distributed inputs. Inputs $s \in \mathcal{N}(\mu, \sigma)$ provide Fisher information $J_s = \frac{1}{\sigma^2}$ about mean $\mu$. Stochastic activity $n(t)$ of a birth-death process ($N \in \{10, 20, 40, 80\}$ and $w = 2.5$) driven by such inputs accumulates Fisher information $J_n(t)$ about mean $\mu$. (a) Accumulation over input interval $t = [0, 1]$ of fractional information $J_{rel}(t) = J_n(t)\,\sigma^2$ by an initially inactive pool of size $N$. (b) Information about μ retained by *summed activity* $\hat{n} = n_1 + \ldots + n_4$ of four independent pools

*Appendix 1—figure 3 continued*

(all initially inactive and of size $N$) receiving concurrently four independent inputs ($s \in \mathcal{N}(\mu, \sigma)$) over an interval $t = [0, 1]$. Retained fraction $J_{rel} = J_{\tilde{n}}(1)\,\sigma^2/4$ depends on pool size $N$ and input variance $\sigma^2$. (**c**) Information about $\mu$ retained by activity $n$ of one pool (initially inactive and of size $N$) receiving successively four independent inputs ($s \in \mathcal{N}(\mu, \sigma)$) over an interval $t = [0, 4]$. Retained fraction $J_{rel} = J_n(4)\,\sigma^2/4$ depends on pool size $N$ and input variance $\sigma^2$.

## Integration of noisy samples

Birth-death processes are able to encode also noisy sensory inputs, capturing much of the information provided. When an initially inactive pool receives an input $s$ over time $t$, stochastic activity $n(t)$ gradually accumulates information about the value of $s$. Normally distributed inputs $s \in N(\mu, \sigma)$ provide Fisher information $J_s = 1/\sigma^2$ about mean μ. Pool activity $n(t)$ accumulates Fisher information $J_n(t)$ about input mean μ, which may be compared to $J_s$. Comparatively small pools with strong coupling (e.g., $N = 25$, $w = 2.5$) readily capture 90% of the information provided (***Appendix 1— figure 5a***).

Moreover, pools readily permit information from multiple independent inputs to be combined over space and/or time. For example, the combined activity of four pools ($N = 25$, $w = 2.5$), which receive concurrently four independent samples, captures approximately 80% of the information provided, and a single pool receiving four samples in succession still retains approximately 60% of the information provided (***Appendix 1—figure 5b,c***). In the latter case, retention is compromised by the 'leaky' nature of stochastic integration. Whether signals are being integrated over space or time, the retained fraction of information is highest for inputs of moderate and larger variance $\sigma^2$ (***Appendix 1—figure 5b,c***). This is because inputs with smaller variance are degraded more severely by the internal noise of a birth-death process (i.e., stochastic activations and inactivations).

## Suitability for inference

Summation of heterogeneous neural responses can be equivalent to Bayesian integration of sensory information [***Beck et al., 2008***; ***Pouget et al., 2013***]. In general, this is the case when response variability is 'Poisson-like' and response tuning differs only multiplicatively [***Ma et al., 2006***; ***Ma et al., 2008***]. We now show that bistable stochastic variables $x_i(t)$, with heterogeneous transition rates $\nu_i^{\pm}(s)$, satisfy these conditions as long as synaptic coupling $w$ is uniform.

Assuming initially inactive variables, $x_i(0) = 0$, incremental responses $x_i(\Delta t)$ after a short interval $\Delta t$ are binomially distributed about mean $\langle x_i(\Delta t) \rangle$, which is approximately

$$\langle x_i(\Delta t) \rangle \approx \Delta t \left. \frac{d\langle x_i \rangle}{dt} \right|_{t=0} = \underbrace{\nu_i \Delta t/2}_{\Phi_i} \; \underbrace{e^{ws/2}}_{f(s)}$$

where $\phi_i = \frac{\nu_i \Delta t}{2}$ reflects (possibly heterogeneous) response tuning and $f(s) = e^{ws/2}$ represents a common response function which depends only on synaptic coupling $w$. The Fisher information, about $s$, of individual responses is

$$J_i(s) = \frac{[\partial_s \langle x_i \rangle]^2}{\langle x_i \rangle \; [1 - \langle x_i \rangle]} \approx \phi_i \frac{f'^2(s)}{f(s)}, \qquad \langle x_i \rangle \ll 1$$

as long as expected activation $\langle x_i \rangle$ is small. The Fisher information of summed responses $\sum_i x_i$ is

$$J_{sum} \approx \frac{\left[ f'^2(s) \sum_i \phi_i \right]^2}{f(s) \sum_i \phi_i} = \frac{f'^2(s)}{f(s)} \sum_i \phi_i = \sum_i J_i(s)$$

and equals the combined Fisher information of individual responses. Accordingly, the summation of bistable activities with heterogeneous transition rates $\nu_i$ optimally integrates information, provided expected activations remain small, $\langle x_i \rangle 1$, and synaptic coupling $w$ is uniform.

## Categorical choice

The 'biased competition' circuit proposed here expresses a categorical decision by either raising $r$ towards unity (and lowering $r'$ towards zero) or vice versa. Here, we describe its stochastic steady-state response to constant visual inputs $I = f(c)$ and $I' = f(c')$ and for arbitrary initial conditions of $e$, $e'$, $r$ and $r'$ (**Appendix 1—figure 3**). Note that, for purposes of this analysis, evidence activity $e$, $e'$ was *not* subject to feedback suppression.

The choice is random when the input is ambiguous, $I \simeq I'$, but quickly becomes deterministic with growing input bias $|I - I'| > 0$. Importantly, the choice is consistently determined by visual input for all initial conditions. The 75% performance level is reached for biases $|I - I'| \approx 0.04 \text{ to } 0.06$.

Mutual inhibition $w_{comp}$ controls the width of the ambiguous region around $I = I'$, and self-excitation $w_{coop}$ ensures a categorical decision even for small $I, I' \simeq 0$. The balance between feedforward excitation $w_{exc}$ and inhibition $w_{inh}$ eliminates decision failures for all but the largest values of $I, I' > 0.7$ and reduces the degree to which sensitivity to differential input $|I - I'|$ varies with total input $I + I'$.

For particularly high values of input $I, I' > 0.7$, no categorical decision is reached and activities of both $r$ and $r'$ grow above 0.5. In the full model, such inconclusive outcomes are eliminated by feedback suppression.

## Deterministic dynamics

In the deterministic limit of $N \to \infty$, fractional pool activity $x$ equals its expectation $\langle x \rangle$ and the relaxation dynamics of **Equation 2** becomes

$$\tau_x \frac{dx}{dt} = -x + x_\infty$$

with characteristic time $\tau_x = \frac{1}{\nu^+ + \nu_-} = \Upsilon(\Delta u)$ and asymptotic values $x_\infty = \frac{\nu^+}{\nu^+ + \nu^-} = \Phi(\Delta u)$, where $\Delta u$ is the potential difference. Input dependencies of characteristic time and of asymptotic value follow from **Equation 1**:

$$\Upsilon(s) = \frac{1}{\nu} \text{sech} \frac{\Delta u}{2}, \qquad \Phi(s) = \left[ 1 + e^{-\Delta u} \right]^{-1}$$

### Evidence pools

The relaxation dynamics of evidence pools is given by **Equation 2** and **Equation 3'**. As shown in the next section, reversals occur when evidence difference $|e - e'|$ reaches a reversal threshold $\Delta_{rev}$. For example, a dominance period of evidence $e'$ begins with $e'_{start} = e_{start} + \Delta_{rev}$ and ends when the concurrent habituation of $e'$ and recovery of $e$ have inverted the situation to $e_{end} = e'_{end} + \Delta_{rev}$ (**Appendix 1—figures 4**). Once the deterministic limit has settled into a limit cycle, all dominance periods start from, and end at, the same evidence levels.

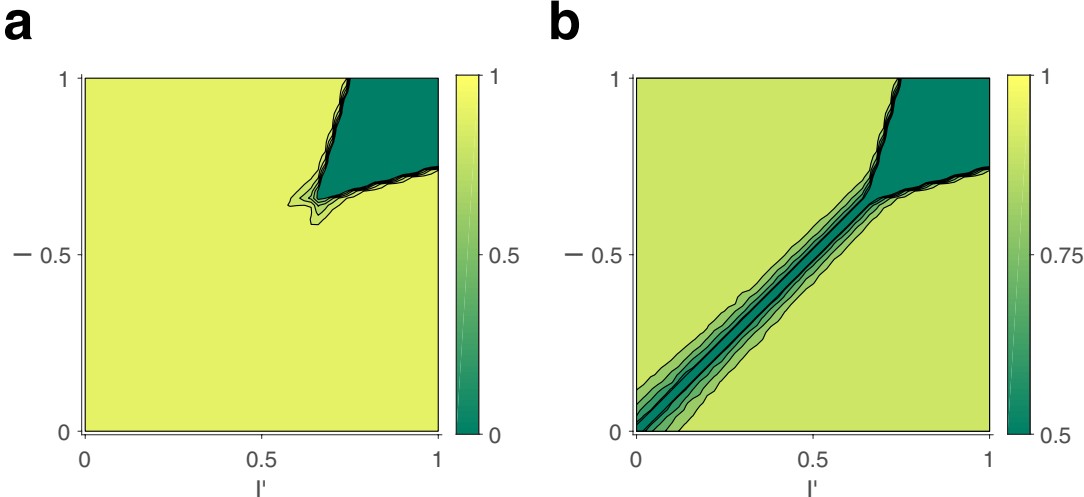

**Appendix 1—figure 4.** Decision response to fixed input $I$, $I'$, for random initial conditions of $e$, $e'$, $r$, $r'$. (a) Expected differential steady-state activation $\langle |r - r'| \rangle$ of decision level. Steady-state activity $r + r' \simeq 1$ implies a categorical decision with activity $\simeq 1$ of one pool and activity of another. (b) Probability that decision correctly reflects input bias ($r > r'$ if $I > I'$), and vice versa.

If pool $R'$ has just become dominant, so that $r' \simeq 1$ and $r \simeq 0$, the state-dependent potential differences are

$$u_e \simeq w_{vis} f(c)$$
$$u_{e'} \simeq w_{vis} f(c') - w_{coop}$$

and the deterministic development is

$$\tau_e \frac{de}{dt} = -e + e_\infty, \qquad \tau_{e'} \frac{de'}{dt} = -e' + e'_\infty$$

with asymptotic values

$$e_\infty = \Phi(u_e + u_e^0), \qquad e'_\infty = \Phi(u_{e'} + u_e^0)$$

and characteristic times

$$\tau_e = \Upsilon\left(u_e + u_e^0\right), \qquad \tau_{e'} = \Upsilon\left(u_{e'} + u_e^0\right),$$

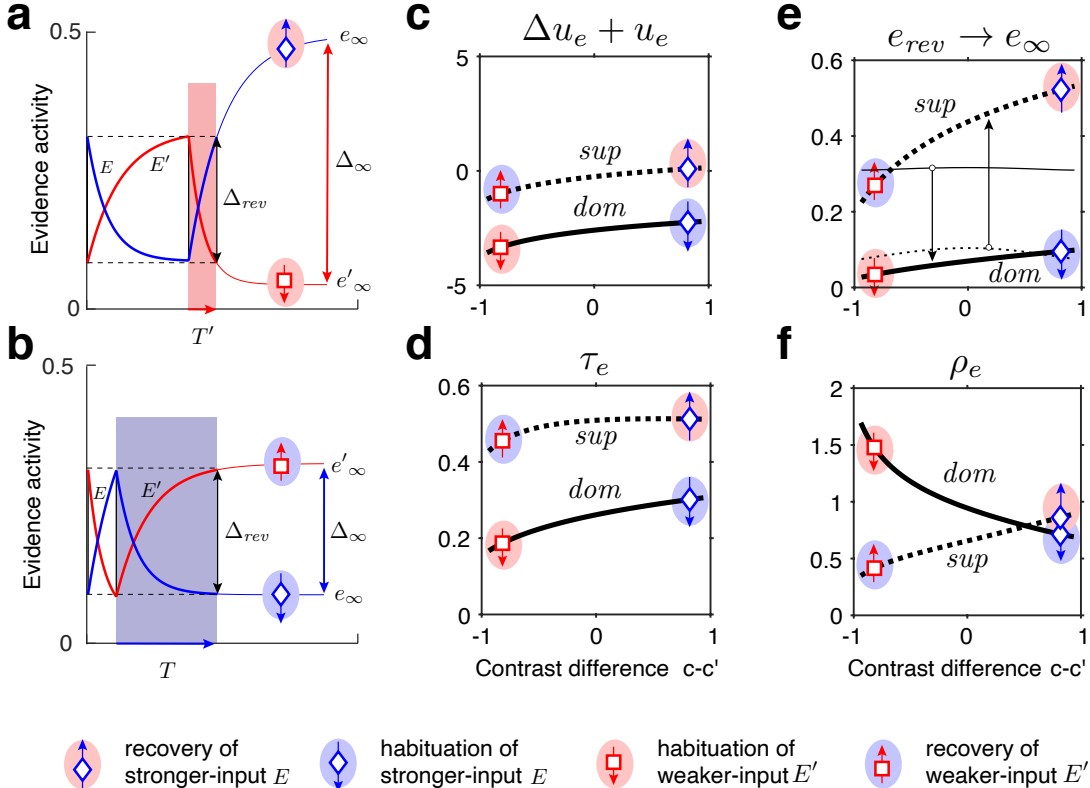

**recovery of stronger-input $E$**

**habituation of stronger-input $E$**

**habituation of weaker-input $E'$**

**recovery of weaker-input $E'$**

**Appendix 1—figure 5.** Exponential habituation and recovery of evidence activities. Dominance durations depend on distance between asymptotic values and on characteristic times. (**a, b**) Development of evidence $e$ (blue) and $e'$ (red), over two successive dominance periods. Input $c = 15/16$ is stronger, input $c' = 1/16$ weaker. Activities recover, or habituate, exponentially until reversal threshold $\Delta_{rev}$ is reached. Thin curves extrapolate to the respective asymptotic values, $e_\infty$ and $e'_\infty$. (**a**) Evidence $e'$ (with weaker input $c'$) is dominant. Incrementing input $c$ to non-dominant evidence $e$ shortens dominance $T'$. (**b**) Evidence $e$ (with stronger input $c$) is dominant. Incrementing input $c$ to $e$ extends dominance $T$. (**c–f**) Contrast dependence of relaxation dynamics, as a function of differential contrast $c - c'$, for $c + c' = 1$. Values when evidence $e$ is dominant (*dom*, thick solid curves), and when it is non-dominant (*sup*, thick dotted curves). Values for $e'$ are mirror symmetric (about vertical midline $c - c' = 0$). (**c**) Effective potential $\Delta u_e + u_e$. (**d**) Characteristic time $\tau_e$. (**e**) Relaxation range $e_{rev} \to e_\infty$ (bottom left, thin curves $e_{rev}$, thick curves $e_\infty$). (**f**) Effective rate $\rho_e$ of development. Symbols and arrows correspond to subfigures (**a, b**) and represent recovery (up arrow) or habituation (down arrow) of stronger-input evidence (blue) or weaker-input evidence (red). Underlying color patches indicate dominance of stronger-input evidence (blue patches) or of weaker-input evidence (red patches). Dominance durations depend more sensitively on the slower development, with smaller $\rho$, which generally is the recovery of non-dominant evidence (up arrows).

The starting points of the development, $e_{rev}$ and $e'_{rev}$ (dashed lines in *Appendix 1—figure 4a,b*), depend mostly on total input $c + c'$ and only little on input difference $c - c'$. Accordingly, for a given level of total input $c + c'$, the situation is governed by the distance between asymptotic evidence levels $\Delta_\infty = e'_\infty - e_\infty$ and by characteristic times $\tau_e, \tau_{e'}$.

The dependence on input bias $c - c'$ of effective potential $\Delta u_e + u_e$, characteristic time $\tau_e$, and asymptotic value $e_\infty$ is illustrated in *Appendix 1—figure 4c–e*. The potential range of relaxation is $e_{rev} \to e_\infty$ and $e'_{rev} \to e'_\infty$, where reversal levels $e_{rev}$ and $e'_{rev}$ can be obtained numerically.

Dominance durations depend more sensitively on the slower of the two concurrent processes as it sets the pace of the combined development. The initial rates $\rho_e$ and $\rho_{e'}$ after a reversal of the two opponent relaxations

$$\rho_e = \frac{d|e|}{dt} = \frac{|e_{rev} - e_\infty|}{\tau_e}, \qquad \rho_{e'} = \frac{d|e'|}{dt} = \frac{|e'_{rev} - e'_\infty|}{\tau_{e'}}$$

provide a convenient proxy for relative rate. As shown in app. *Figure 4f*, when stronger-input evidence $e$ dominates, recovery of weaker-input evidence (red up arrow on blue background) is slower than habituation of stronger-input evidence (blue down arrow on blue background). Conversely, when weaker-input evidence $e'$ dominates, recovery of stronger-input evidence (blue up arrow on red background) is slower than habituation of weaker-input evidence (red down arrow on red background). In short, dominance durations always depend more sensitively on the *recovery* of the currently non-dominant evidence than on the *habituation* of the currently dominant evidence.

If the two evidence populations $E$, $E'$ have equal and opposite potential differences, $\Delta u_e = -\Delta u_{e'}$, then they also have equal and opposite activation and inactivation rates (*Equation 1*)

$$\nu_e^+ = \nu_{e'}^- = \nu^+, \qquad \nu_e^- = \nu_{e'}^+ = \nu^-$$

and identical characteristic times $\tau_e$ (recovery of $E$) and $\tau_{e'}$ (habituation of $E'$). In this special case, the two processes may be combined and the development of evidence difference $\Delta e = e - e'$ is

$$\tau_\Delta \frac{d\Delta e(t)}{dt} = -\Delta e(t) + \Delta_\infty$$

$$\tau_\Delta = \frac{1}{\nu^+ + \nu^-}, \qquad \Delta_\infty = \frac{\nu^+ - \nu^-}{\nu_+ + \nu^-}$$

Starting from $\Delta e(0) = -\Delta_{rev}$, we consider the first-passage-time of $\Delta e(t)$ through $+\Delta_{rev}$. If a crossing is certain (i.e. when $\frac{\nu^+}{\nu^+ + \nu^-} > \Delta_{rev}$), the first-passage-time $T$ writes

$$T = \tau_\Delta \ln\left(\frac{\Delta_\infty + \Delta_{rev}}{\Delta_\infty - \Delta_{rev}}\right) \tag{2}$$

A similar hyperbolic dependence obtains also in all other cases. When the distance between asymptotic levels $\Delta_\infty$ falls below the reversal threshold $\Delta_{rev}$, dominance durations become infinite and reversals cease.

The hyperbolic dependence of dominance durations, illustrated in *Appendix 1—figure 4d*, has an interesting implication. Consider the point of equidominance, at which both dominance durations are equal and of moderate duration. Increasing the difference between image contrasts (e.g., increasing $c \to c + \Delta c$ and decreasing $c' \to c' - \Delta c$) increases $\Delta_\infty$ during the dominance of $e$ and decreases it during the dominance of $e'$. Due to the hyperbolic dependence, longer dominance periods lengthen more ($T \to T + \Delta T$) than shorter dominance periods shorten ($T' \to T' - \Delta T'$), consistent with the contemporary formulation of Levelt III [*Brascamp et al., 2015*].

### Decision pools

We wish to analyze steady-state conditions for decision pools $R$, $R'$, as illustrated in *Appendix 1—figure 4a,b*. From *Equation 4*, we can write

$$r_\infty = \phi\left(e, e', r_\infty, r'\right), \qquad r'_\infty = \phi\left(e, e', r, r'_\infty\right),$$

Under certain conditions – in particular, for sufficient self-coupling $w_{coop}$ – the steady-state equations admit more than one solution: a low-activity fixed point with $r'_\infty \simeq 0$, and a high-activity fixed point with $r_\infty \simeq 1$. Importantly, the low-activity fixed point can be destabilized when evidence activities change.

Consider a non-dominant decision pool $R$ with fractional activity $r = n_r/N \simeq 0$ and its dominant rival pool $R'$ with fractional activity $r' = n_{r'}/N \simeq 1$. The steady-state condition then becomes

$$r_\infty \simeq \phi\left[w_{coop}\left(r_\infty - x_{eff}\right)\right]$$
$$x_{eff} = \frac{w_{comp} - w_{exc}e + w_{inh}(e + e') - u_r}{w_{coop}}$$

For certain values $x_{eff} \leq x_{crit}$, the low-activity fixed point becomes unstable, causing a sudden upward activation of pool $R$ and eventually a perceptual reversal. We call $r_{crit}$ the steady-state value of $r_\infty$ at the point of disappearance.

We can now define a threshold $\Delta_{rev}$ in terms of the value of evidence bias $\Delta e = e - e'$ which ensures that $x_{eff} \geq x_{crit}$:

$$\Delta_{rev} \geq \frac{2}{w_{exc}}\left(w_{comp} - x_{crit}w_{coop} - u_r\right) - \frac{2}{w_{exc}}(w_{exc} - 2w_{inh})\frac{e + e'}{2}$$

We find that the threshold value $\Delta_{rev}$ decreases linearly with average evidence $\bar{e} = (e + e')/2$, so that higher evidence activity necessarily entails lower thresholds (dashed red line in *Figure 5c*).

For $w_{coop} = 15.21$, we find $x_{crit} = 0.24006$, $r_{crit} = 0.0708$, and $\Delta_{rev} = 0.4554 - 1.1564\,\bar{e}$.

**Appendix 1—figure 6.** Birth-death dynamics of evidence pools ensures gamma-like distribution and 'scaling property' (invariance of distribution shape). (**a**) Representative examples for the time development of evidence bias $\Delta e = e - e'$ between reversals (i.e., between $-\Delta_{rev}$ and approximately $+\Delta_{rev}$). (**b**) Dominance distributions for $c = c' = 1/16$ (blue), $c = c' = 1/4$ (green), and $c = c' = 1$ (yellow). Distribution mean $\mu$ changes approximately threefold, but coefficient of variation $c_V$ and skewness $\gamma_1$ are nearly invariant (inset), largely preserving distribution shape. (c) Development of expectation $\langle\Delta x\rangle$ between reversals (schematic). Left: a Poisson variable process, such

*Appendix 1—figure 6 continued on next page*

*Appendix 1—figure 6 continued*

as the difference $\Delta x$ between two birth-death processes. Mean $\langle \Delta x \rangle$ grows linearly with $t$ (lines, with slopes $\mu$, $\mu'$) and variance $\langle (\Delta x - \langle \Delta x \rangle)^2 \rangle$ grows linearly with $\sqrt{t}$ (dashed curves, with scaling factors $\sigma$, $\sigma'$). Constants $\mu$ and $\sigma$ change with stimulus contrast (blue and red). Proportionality $\mu \propto \sigma^2$ ensures constant dispersion of $\Delta x$ at threshold ($\delta_x = \delta'_x$), and, consequently, a dispersion of threshold-crossing times that grows linearly with mean threshold-crossing time ($\delta_t / t_{rev} = \delta'_t / t'_{rev} = const$), preserving distribution shape. Right: a process with constant variance, $\sigma = \sigma'$. Dispersion of $\Delta x$ at threshold increases with threshold-crossing time ($\delta'_x > \delta_x$) and dispersion of threshold-crossing times grows supra-linearly with mean threshold-crossing time ($\delta_t / t_{rev} < \delta'_t / t_{rev} < \delta'_t / t'_{rev}$), broadening distribution shape.

## Potential landscape

In **Figure 5b**, we illustrate the steady-state condition $r_\infty = \phi \left[ w_{coop} \left( r_\infty - x_{eff} \right) \right]$ in terms of an effective potential landscape $U(x)$. The functional form of this landscape was obtained by integrating 'restoring force' $F(x)$ over activity $x$:

$$F(x) = \Phi[w_{coop}(x - x_{eff})] - x,$$
$$u(x) = - \oint_{x_{eff}}^{x} F(x')dx'$$

## **Stochastic dynamics**

### Poisson-like variability

The discretely stochastic process $x(t) \in \{0, \frac{1}{N}, \frac{2}{N}, ..., 1\}$ has a continuously stochastic 'diffusion limit,' $x_{diff}(t)$, for $N \to \infty$, with identical mean $\langle x_{diff} \rangle = \langle x \rangle$ and variance $\langle x_{diff}^2 \rangle - \langle x_{diff} \rangle^2 = \langle x^2 \rangle - \langle x \rangle^2$. This diffusion limit is a Cox–Ingersoll process and its dynamical equation

$$\dot{x}_{diff} = \left( 1 - x_{diff} \right) \nu^+ - x_{diff} \nu^- + \sqrt{\frac{\left( 1 - x_{diff} \right) \nu^+ + x_{diff} \nu^-}{N}} \, \xi(t),$$

where $\xi(t)$ is white noise, reveals that its increments $N \dot{x}_{diff}$ (and thus also the increments of the original discrete process) exhibit Poisson-like variability. Specifically, in the low-activity regime, $x_{diff} \ll 1$, both mean and variance of increments approximate activation rate $N \nu^+$:

$$\langle N\dot{x} \rangle = \langle N\dot{x}_{diff} \rangle \simeq N\nu^+,$$
$$\langle N^2 \dot{x}^2 \rangle - \langle N\dot{x} \rangle^2 = \langle N^2 \dot{x}_{diff}^2 \rangle - \langle N\dot{x}_{diff} \rangle^2 \simeq N\nu^+.$$

### Gamma-distributed first-passage times

When the input to a pool of bistable variables undergoes a step change, the active fraction $x(t)$ transitions stochastically between old and new steady states, $x_\infty^{old}$ and $x_\infty^{new}$ (set by old and new input values, respectively). The time that elapses until fractional activity crosses an intermediate 'threshold' level $\theta$ ($x_\infty^{old} < \theta < x_\infty^{new}$) is termed a 'first-passage-time.' In a low-threshold regime, birth-death processes exhibit a particular and highly unusual distribution of first-passage times.

Specifically, the distribution of first-passage-times assumes a characteristic, gamma-like shape for a wide range of value triplets ($x_\infty^{old}$, $\theta$, $x_\infty^{new}$) [**Cao et al., 2014**]: skewness $\gamma_1$ takes a stereotypical value $\gamma_1 \simeq 2c_V$, the coefficient of variation $c_V$ remains constant (as long as the distance between $x_\infty^{old}$ and $\theta$ remains the same), whereas the distribution mean may assume widely different values. This gamma-like distribution shape is maintained even when shared input changes during the transition (e.g., when bistable variables are coupled to each other) [**Cao et al., 2014**].

Importantly, only a birth-death process (e.g., a pool of bistable variables) guarantees a gamma-like distribution of first-passage-times under different input conditions [25]. Many other discretely stochastic processes (e.g., Poisson process) and continuously stochastic processes (e.g., Wiener, Ornstein–Uhlenbeck, Cox–Ingersoll) produce inverse Gaussian distributions with $\gamma_1 \simeq 3 c_V$. Models combining competition, adaptation, and noise can produce gamma-like distributions, but require different parameter values for every input condition (see Materials and methods: *Alternative model*).

## Scaling property

In the present model, first-passage-times reflect the concurrent development of two opponent birth-death processes (pools of $N = 25$ binary variables). Dominance periods begin with newly non-dominant evidence $e$ *well below* newly dominant evidence $e'$, $\Delta e = e - e' \simeq -\Delta_{rev}$, and end with the former *well above* the latter, $\Delta e \simeq +\Delta_{rev}$ (*Appendix 1—figure 6a*). The combination of two small pools with $N = 25$ approximates a single large pool with $N = 25$. When image contrast changes, distribution shape remains nearly the same, with a coefficient of variation $c_V \simeq 0.6$ and a gamma-like skewness $\gamma_1 \simeq 2$, even though mean $\mu$ of first-passage-times changes substantially (*Appendix 1—figure 6b*).

This 'scaling property' (preservation of distribution shape) is owed to the Poisson-like variability of birth-death processes (see above, *Appendix 1—figure 6c*). Poisson-like variability implies that accumulation rate $\mu$ and dispersion rate $\sigma^2$ are proportional, $\mu \sim \sigma^2$. This proportionality ensures that activity at threshold disperses equally widely for different accumulation rates (i.e., for different input strengths), preserving the shape of first-passage-time distributions [*Cao et al., 2016*].

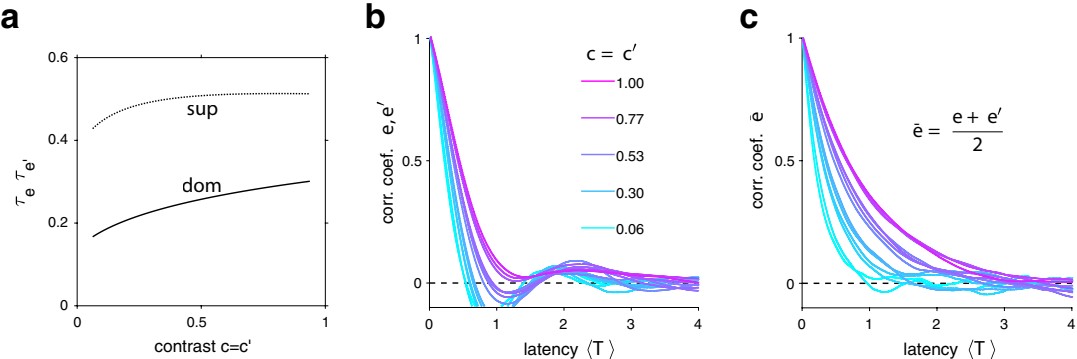

**Appendix 1—figure 7.** Characteristic times of evidence activity. (**a**) Characteristic times $\tau_e$, $\tau_{e'}$ for different image contrast $c = c'$, when evidence pool is dominant (*dom*) and non-dominant (*sup*). (**b**) Autocorrelation of evidence activity $e$, $e'$ as a function of image contrast (color) and latency, expressed in multiples of average dominance duration $\langle T \rangle$. (**c**) Autocorrelation of joint evidence activity $\bar{e} = (e + e')/2$ as a function of image contrast (color) and latency. Note that autocorrelation time lengthens substantially for high image contrast.

## Characteristic times

As mentioned previously, the characteristic times of pools of bistable variables are not fixed but vary with input (*Equation 2*). In our model, the characteristic times of evidence activities lengthen with increasing input contrast and shorten with feedback suppression (*Appendix 1—figure 7a*). Characteristic times are reflected also in the temporal autocorrelation, which averages over periods of dominance and non-dominance alike. Autocorrelation times lengthen with increasing input contrast, both in absolute terms and relative to the average dominance duration (*Appendix 1—figure 7b*).

Importantly, the autocorrelation time of *mean* evidence activity $\bar{e} = (e + e')/2$ is even longer, particularly for high input contrast (*Appendix 1—figure 7c*). The reason is that spontaneous fluctuations of $\bar{e}$ are constrained not only by birth-death dynamics, but additionally by the reversal dynamics that keeps evidence activities $e$ and $e'$ close together (i.e., within reversal threshold $\Delta_{rev}$). As a result, the characteristic timescale of spontaneous fluctuations of $\bar{e}$ lengthens with input contrast. The amplitude of such fluctuations also grows with contrast (not shown).

The slow fluctuations of $\bar{e}$ induce mirror-image fluctuations of reversal threshold $\Delta_{rev}$ and thus are responsible for the serial dependency of reversal sequences (see Deterministic dynamics: *Decision pools*).

## Burstiness

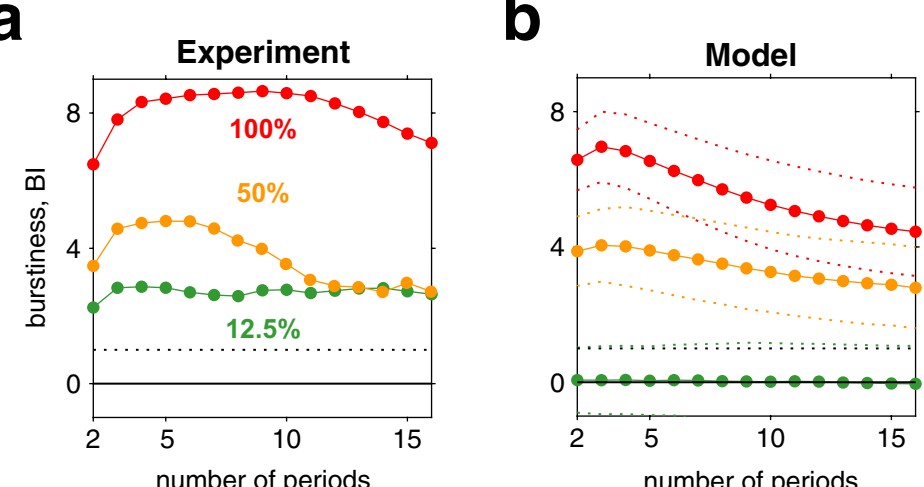

**Appendix 1—figure 8.** Burstiness of reversal sequences predicted by model and confirmed by experimental observations. (**a**) Burstiness index (BI) (mean) for $n$ successive dominance periods in experimentally observed reversal sequences, for contrasts 12.5% (green), 50% (yellow), and 100% (red). (**b**) BI for reversal sequences generated by model (mean ± std).

The proposed mechanism predicts that reversal sequences include episodes with several successive short (or long) dominance periods. It further predicts that this inhomogeneity increases with image contrast. Such an inhomogeneity may be quantified in terms of a 'burstiness index' (BI), which compares the variability of the mean for sets of $n$ successive periods to the expected variability for randomly shuffled reversal sequences. In both model and experimental observations, this index rises far above chance (over broad range of $n$) for high image contrast (*Appendix 1—figure 8*). The degree of inhomogeneity expressed by the model at high image contrast is comparable to that observed experimentally, even though the model was neither designed nor fitted to reproduce non-stationary aspects of reversal dynamics. This correspondence between model and experimental observation compellingly corroborates the proposed mechanism.

## Robustness of fit

The parameter values associated with the global minimum of the fit error define the model used throughout the article. As described in Materials and methods, we explored the vicinity of this parameter set by individually varying each parameter within a certain neighborhood. This allowed us to estimate 95% confidence intervals for each parameter value. The results are illustrated in *Appendix 1—figure 9*.

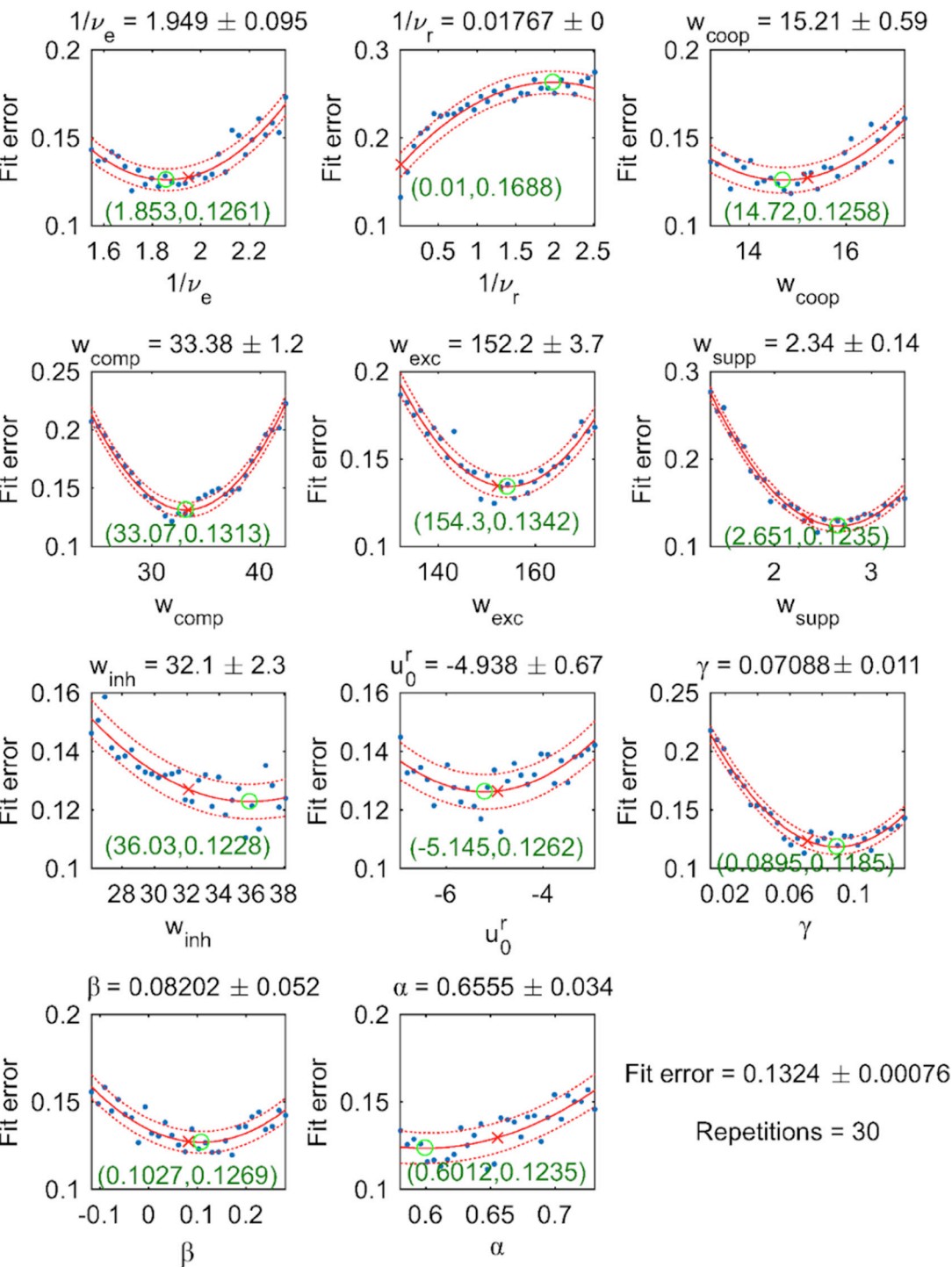

**Appendix 1—figure 9.** Dependence of fit error on individual parameter values (with all other parameter values fixed). 30 equally spaced values were tested (blue dots) and fitted by a quadratic function (red solid curve, with 95% confidence intervals indicated by dotted curves). For each parameter, both the optimal value (red cross) and the extremum of the parabolic fit (green circle) are shown.

Note that optimal parameter values (red crosses) are consistently near extrema of the parabolic fits (green circles), indicating the robustness of the fit. Note further that instead of the parameter pair $w_{vis}$ and $u_e^0$, we show the related parameter pair $\alpha$ and $\beta$, which is defined through the relations $w_{vis} = \alpha \ln((1 + \gamma)/\gamma)$ and $u_e^0 = \alpha \ln \gamma + \beta$.

The code used to analyze optimization statistics is available in the folder 'analyzeOptimizationStatistics' of the Github repository provided with this article (https://github.com/mauriziomattia/2021.Bistable PerceptionModel) copy archived at.

