## [Decision Letter]

**Acceptance summary:**

In a computational model of perceptual bistability, here binocular rivalry, a novel hierarchical structure includes an evidence accumulation stage and a competition stage that together support continuous decision-making and perceptual alternations, and account for the statistics of percept durations and serial dependencies, as well as Levelt's four propositions in the study's behavioral data. Of particular interest, feedback inhibition from the perceptual competition to the sensory evidence accumulation stage provides a gating mechanism on sensory units so that decision-making is based on evidence against the current percept.

**Decision letter after peer review:**

Thank you for submitting your article "Instability of visual perception reveals the dynamics of decision-making in a volatile world" for consideration by *eLife*. Your article has been reviewed by 2 peer reviewers, and the evaluation has been overseen by a Reviewing Editor and Joshua Gold as the Senior Editor. The following individuals involved in review of your submission have agreed to reveal their identity: James Rankin (Reviewer #1); Chris Klink (Reviewer #2).

The reviewers have discussed the reviews with one another and the Reviewing Editor has drafted this decision to help you prepare a revised submission.

We would like to draw your attention to changes in our revision policy that we have made in response to COVID-19 (https://elifesciences.org/articles/57162). Specifically, we are asking editors to accept without delay manuscripts, like yours, that they judge can stand as *eLife* papers without additional data, even if they feel that they would make the manuscript stronger. Thus the revisions requested below address clarity and presentation, including some toning down of over-generalization.

This paper presents valuable advances to the modeling of perceptual bistability, here applied to binocular rivalry. The novel hierarchical structure includes an evidence accumulation stage and a competition stage that together support continuous decision-making and perceptual alternations. Importantly, the study is used to account for the statistics of percept durations and as well as Levelt's four propositions in the behavioral data.

Summary:

This paper presents an original hierarchical model for the perceptual dynamics of binocular rivalry. The model separates a stage of perceptual evidence accumulation from a stage of dominance competition and implements feedback inhibition from the competition to the evidence accumulation stage. This feedback provides a gating mechanism on sensory units so that decision-making is based on evidence against the current percept. The study demonstrates a potential neural mechanism that builds on previous models of decision-making and rivalry. Decision making in the model is continuous and leads to alternations that mimic statistical properties of behavioral perceptual bistability, including correlations from percept to percept. The authors effectively extract explanatory value from their model and its mechanisms, using analytic approximations based on statistical dynamics.

The study brings together, in the context of binocular rivalry, two existing strands of modelling perceptual bistability: capturing the statistical properties of dominance durations (a universal distribution shape independent of stimulus parameters, correlations etc) and capturing their dependence on input strengths (Levelt's propositions for binocular rivalry). The results presented go substantially beyond a long-established modelling literature.

The paper is well written and the figures clearly presented. The core results are described in a suitable style so as to keep the paper accessible to a wide readership. The later and supporting figures become more technical but this is necessary to draw out a deeper understanding as to why the model works.

We offer the following suggestions to improve the manuscript.

Essential revisions:

1. The presentation has a tone of over-generalized claims. Please keep closer to the Results – to what has been done. For example, statements about normative constraints and volatile world can be toned down. The paper does not directly assess the model's performance or auto-adjustment in dynamic and unpredictable ('volatile') environments as would be with normative modeling. Such statements do not reflect primary results and should be restricted primarily to the Discussion. They should not be in the paper's title. Please have the title refer to Binocular Rivalry. Also note: the journal *eLife* discourages 2-part titles with ":".

Additional data/experiments are not required if the authors drop the over-generalizations. However, if they do want to make more general claims, these should be backed up.

2. Re: Robustness of the simulation results. Some indication of model robustness should be provided to demonstrate whether all these nicely fitting dynamics do crucially depend on a precise set of parameters. Presumably in different parameter regions the model could also produce negative lag-2 correlations. Is it not just luck that the best-fit fell in a region with positive lag-2 correlations as in the data. On the other hand, are the authors able to show that if the lag-2 data is included in the fit function for the alternative model it is incapable of producing the positive correlations? These issues could be addressed in the Discussion.

Additional data/experiments are not required if the authors drop the over-generalizations. However, if they do want to make more general claims, these should be backed up..

3. Re: perceptual bistability is restricted here to binocular rivalry. Topics for the Discussion:

a. The occurrence of piecemeal or mixed percepts are not addressed. In the experiments, such perhaps are identified (with released keys) but they're not analyzed, nor modeled. Instead, bistable assemblies are assumed binary. Phenomenologically, this is definitely not the case and given existing research on mixed percepts, this should at least be discussed.

b. How does the model deal with other known binocular rivalry phenomena like flash suppression, or priming, intermittent presentation, or different time-scales? These are phenomena that were previously modeled with a strong role for adaptation. Are there any unexpected prediction the model makes beyond currently known binocular rivalry dynamics that can be tested experimentally?

c. Can the authors discuss their framework in the context of the visual system and competition between binocular and feature representations of the stimuli (in the vein of Wilson 2003)?

d. Another hierarchical model (Li, H.-H.; Rankin, J.; Rinzel, J.; Carrasco, M. and Heeger, D. Attention model of binocular rivalry P Natl Acad Sci, 2017, 114), building on the framework proposed by the cited Wilson 2003 study, proposes a descending feature-based excitation as a proxy for exogenous attention (in contrast with feedback suppression proposed here). Can the authors comment on the features of their data that this alternative mechanism might fail to capture. How could the effects of attention (or its absence) be explained by the model presented here?

4. Modeling aspects:

a. Perceptual bistability for perceptual grouping (as for ambiguous auditory stimuli) has been modeled explicitly as evidence accumulation against the current percept; please discuss and relate your approach to these: Barniv and Nelken (2015), Nguyen, Rinzel and Curtu (2020)

b. The model assumes two sets of independent bistable feature detectors. Please relate this assumption to potential brain areas and evidence that units with similar receptive fields may operate independently, as uncoupled units. (c) What other experiments on decision making (perceptual or otherwise) might this framework be able to speak to?

5. Re: specific conclusions: From one reviewer: Whilst I clearly understand what has been achieved I found it hard to pin down the specific conclusions from this study. State them explicitly, at the end of the discussion? Do not omit more specific conclusions based on this model and binocular rivalry before any other more general conclusions that draw from a wider literature.

6. The authors should post/link their model codes to a well-documented software repository. Interested readers could then see the model in action and inspect what happens internally.

---

## [Author Response]

Essential revisions:1. The presentation has a tone of over-generalized claims. Please keep closer to the Results – to what has been done. For example, statements about normative constraints and volatile world can be toned down. The paper does not directly assess the model's performance or auto-adjustment in dynamic and unpredictable ('volatile') environments as would be with normative modeling. Such statements do not reflect primary results and should be restricted primarily to the Discussion. They should not be in the paper's title. Please have the title refer to Binocular Rivalry. Also note: the journal eLife discourages 2-part titles with ":".Additional data/experiments are not required if the authors drop the over-generalizations. However, if they do want to make more general claims, these should be backed up.

Title and abstract were revised as follows:

**“**Binocular rivalry reveals an out-of equilibrium neuronal dynamics suited for decision-making**”.**

“In ambiguous or conflicting sensory situations, perception is often ‘multistable’ in that it perpetually changes at irregular intervals, shifting abruptly between distinct alternatives. […] Thus, multistable perception may reflect decision-making in a volatile world: integrating evidence over space and time, choosing categorically between hypotheses, while concurrently evaluating alternatives.”

2. Re: Robustness of the simulation results. Some indication of model robustness should be provided to demonstrate whether all these nicely fitting dynamics do crucially depend on a precise set of parameters.

For each parameter value, we now provide confidence intervals (—Appendix—figure 9), which we obtained by establishing the dependence of fit error on individual parameter values (while other parameters remained fixed). Typically, fit error increases parabolically around the optimal parameter value.

This is now described in the new section "Robustness of fit" in the Supplementary Methods.

Red crosses represent the optimal model parameters while green circles are associated to the extrema of the quadratic fit of the computed fit error obtained by varying in a suited range the single eleven parameters. Dotted lines are the 95% confidence intervals of the fit. For each parameter, 30 nearby equally spaced values have been tested. This allowed us to estimate the error of the parameter values found from the original fitting procedure. Note that instead of the parameters wvis and u0 e we show α and β which are directly related parameters: wvis= αln⁡1+γγ and u0 e= α ln γ+ β.

Presumably in different parameter regions the model could also produce negative lag-2 correlations. Is it not just luck that the best-fit fell in a region with positive lag-2 correlations as in the data.

The model consistently predicts positive lag-2 correlations, at least in parameter regimes that reproduce Levelt's propositions (dependence of dominance durations on image contrast). As explained in Results under 'Serial dependence', these correlations result from an interaction between evidence and decision level dynamics.

Specifically, dominance periods reflect the current reversal threshold (defined as the expected value of at reversal time), because a longer (shorter) dominance period is needed to reach a higher (lower) threshold. In turn, reversal threshold tracks the slow fluctuations of combined evidence activity: e¯=(e+e′)/2 more (less) evidence activity entails a lower (higher) reversal thresholdΔrev

As shown in Supplementary Figure 13, the autocorrelation times of e¯ (and ) are substantially longer than mean dominance periods, especially for large image contrasts. Due to this long autocorrelation, both reversal thresholds and dominance durations are strictly *positively* correlated over successive reversals.

On the other hand, are the authors able to show that if the lag-2 data is included in the fit function for the alternative model it is incapable of producing the positive correlations? These issues could be addressed in the Discussion.Additional data/experiments are not required if the authors drop the over-generalizations. However, if they do want to make more general claims, these should be backed up.

The alternative model, which includes two independent adaptive states and ' consistently produced *positive* lag-1 correlations and *negative* lag-2 correlations for all combinations of image contrast fitted to reproduce the moments of dominance distributions (mean, CV, skewness).

In a Laing and Chow-type model, states and ' develop deterministically in an opponent fashion, in that an increase of one state necessarily entails a decrease of the other. Adaptive bias ' represents a negative memory of recent dominance history. At the end of a long period of dominance, from this bias for the new percept. During the new dominance period, bias will diminish but will not necessarily *change direction*. *Negative* lag-2 correlations result when the direction of persists to the next reversal, so that it once again disfavours the old (original) percept.

In our experience, the prediction of positive lag-1 and negative lag-2 correlations is robust and statistically significant, provided sufficiently long reversal sequences are generated. Parenthetically, we note that we gained extensive experience in reproducing moments of observed dominance distributions with Laing and Chow-type models in the context of other studies (102; Ziman et al., 2021: https://www.biorxiv.org/content/10.1101/2021.02.11.430816v1).

In fact, Laing and Chow (2002) predicted positive and negative correlations at lag-1 and lag-2, respectively (see their Figure 5), but chose to simulate only a small number of reversals, so that these correlations failed to reach significance.

However, we do not consider the present paper a good venue for an in depth evaluation of Laing and Chow-type models.

3. Re: perceptual bistability is restricted here to binocular rivalry. Topics for the Discussion:a. The occurrence of piecemeal or mixed percepts are not addressed. In the experiments, such perhaps are identified (with released keys) but they're not analyzed, nor modeled. Instead, bistable assemblies are assumed binary. Phenomenologically, this is definitely not the case and given existing research on mixed percepts, this should at least be discussed.

Discussion, page 25:

“Binocular rivalry arises within local regions of the visual field, measuring approximately 0.25 deg in the fovea [Leopold, 1997; Logothetis, 1998]. […] To account for these phenomena, the visual field would have to be tiled with replicant models linked by grouping interactions [Knapen et al., 2007; Bressloff et al., 2012].”

b. How does the model deal with other known binocular rivalry phenomena like flash suppression, or priming, intermittent presentation, or different time-scales? These are phenomena that were previously modeled with a strong role for adaptation.

In response, the discussion was extended by the following sections:

Discussion, page 24:

“Unlike many previous models [e.g. Laing and Chow, 2002; Wilson, 2007; Moreno-Bote et al., 2007; Moreno-Bote et al., 2010; Cohen et al., 2019], the proposed mechanism does not include adaptation (stimulation-driven weakening of evidence), but a phenomenologically similar feedback suppression (perception-driven weakening of evidence). […] The extent to which a perception-driven suppression could also explain these observations remains an open question for future work.”

Discussion, page 25:

“Multistable perception induces a positive priming or ‘sensory memory’ [Pearson et al. 2005; Pastukhov et al., 2008, Pastukhov et al., 2013a], which can stabilize a dominant appearance during intermittent presentation [Leopold et al., 2003; Maier et al., 2003; Sandberg et al., 2014]. This positive priming exhibits rather different characteristics (e.g., shape-, size- and motion- specificity, inducement period, persistence period) than the negative priming / adaptation of rivalling representations [De Jong et al., 2012; Pastukhov et al., 2013a, 2013b, 2014a, 2014b, 2016]. […] To incorporate sensory memory, the present model would have to be extended to include three hierarchical levels (evidence, decision, and memory), as previously proposed by [Gigante et al., 2009].”

Are there any unexpected prediction the model makes beyond currently known binocular rivalry dynamics that can be tested experimentally?

Due to a highly non-trivial interaction with the competitive decision, discretely stochastic fluctuations of evidence level activity express themselves in a serial dependency of dominance durations. Several features of this dependency were unexpected and not reported previously, for example, the sensitivity to image contrast and also the `burstiness' of dominance reversals (i.e., extended episodes in which dominance periods are consistently longer or shorter than average). The fact that these predictions are confirmed by our experimental observations supports the proposed mechanism.

In addition, the hierarchical and modular architecture of the present model lends itself to a number of extensions, for example, to multistable phenomena with more than two competing appearances, or to situations with volatile and unpredictable input. We are currently working to develop additional predictions that we believe to be beyond the scope of the present manuscript.

c. Can the authors discuss their framework in the context of the visual system and competition between binocular and feature representations of the stimuli (in the vein of Wilson 2003)?

Discussion, page 25:

“A particularly intriguing previous model [Wilson, 2003] postulated a hierarchy with competing and adapting representations at two separate levels (8 state variables), one lower (monocular) and another higher (binocular) level. […] It is tempting to speculate that such ‘stacking’ might have a normative justification in that it might subserve hierarchical inference [Yuille et al., 2006; Hohwy and Friston, 2008, Friston, 2010].”

d. Another hierarchical model (Li, H.-H.; Rankin, J.; Rinzel, J.; Carrasco, M. and Heeger, D. Attention model of binocular rivalry P Natl Acad Sci, 2017, 114.), building on the framework proposed by the cited Wilson 2003 study, proposes a descending feature-based excitation as a proxy for exogenous attention (in contrast with feedback suppression proposed here). Can the authors comment on the features of their data that this alternative mechanism might fail to capture. How could the effects of attention (or its absence) be explained by the model presented here?

Discussion, page 25/26:

“Another previous model [Li et al., 2017] used a hierarchy with three separate levels (and 24 state variables) to show that a stabilizing influence of selective visual attention could also explain slow rivalry when images are swapped rapidly. […] As the present model already incorporates the ‘biased competition’ that is widely thought to underlie selective attention [Kastner et al., 2000; Reynolds and Heeger, 2009], we expect that it could reproduce attentional effects by means of additive modulations.”

4. Modeling aspects:a. Perceptual bistability for perceptual grouping (as for ambiguous auditory stimuli) has been modeled explicitly as evidence accumulation against the current percept; please discuss and relate your approach to these: Barniv and Nelken (2015), Nguyen, Rinzel and Curtu (2020)

Discussion, page 26/27:

“Continuous inference has been studied extensively in auditory streaming paradigms [Winkler et al., 2012; Denham et al., 2014]. […] The dynamics of two recent auditory models [Barniv and Nelken, 2015; Nguyen et al., 2020] are rather similar to the model presented here: while one sound pattern dominates awareness, evidence against this pattern is accumulated at a subliminal level.”

b. The model assumes two sets of independent bistable feature detectors. Please relate this assumption to potential brain areas and evidence that units with similar receptive fields may operate independently, as uncoupled units.

Discussion, page 27/28:

“Unfortunately, the spatial structure of the "shared variability" and "noise correlations" described above is poorly understood. […] Additionally, a new section was added on the neurophysiology of rivalry and its relation to our model.”

Discussion, page 27/28:

“Neurophysiological correlates of binocular rivalry

Neurophysiological correlates of binocular rivalry have been studied extensively, often by comparing reversals of phenomenal appearance during binocular stimulation with physical alternation of monocular stimulation [e.g., Leopold and Logothetis, 1996; Scheinberg and Logothetis, 1997; Wilke et al., 2006; Maier et al., 2008; Keliris et al., 2010; Panagiotaropoulos et al., 2012; Bahmani et al., 2014; Xu et al., 2016; Kapoor et al., 2020; Dwarakanath et al., 2020]. At higher cortical levels, such as inferior temporal cortex [Scheinbert and Logothetis, 1997] or prefrontal cortex [Panagiotaropoulos et al., 2012; Kapoor et al., 2020; Dwarakanth et al., 2020], binocular rivalry (BR) and physical alternation (PA) elicit broadly comparable neurophysiological responses that mirror perceptual appearance. […] Presumably, a “stacked” model with two successive levels of competitive interactions at monocular and binocular levels or representation [Wilson, 2003; Li et al., 2017] would be needed to account for these phenomena.”

c. What other experiments on decision making (perceptual or otherwise) might this framework be able to speak to?

The hierarchical and modular architecture of the present model lends itself to a number of extensions, for example, to multistable phenomena with more than two competing appearances, or to continuous detection of alternative stimuli in volatile and unpredictable environments. We are currently working to develop additional predictions that we believe to be beyond the scope of the present manuscript.

5. Re: specific conclusions: From one reviewer: Whilst I clearly understand what has been achieved I found it hard to pin down the specific conclusions from this study. State them explicitly, at the end of the discussion? Do not omit more specific conclusions based on this model and binocular rivalry before any other more general conclusions that draw from a wider literature.

Discussion, page 22:

“We have shown that many well-known features of binocular rivalry are reproduced, and indeed guaranteed, by a particular dynamical mechanism. […] Indeed, the phenomena explained exhibited more effective degrees of freedom (approximately 14) than the mechanism itself (between 3 and 4).”

6. The authors should post/link their model codes to a well-documented software repository. Interested readers could then see the model in action and inspect what happens internally.

We are of course happy to make available Matlab code for interested readers and have created a GitHub repository for this purpose: https://github.com/mauriziomattia/2021.BistablePerceptionModel

The folder 'plotExampleNetworkDynamics' generates and illustrates realizations of the hierarchical model dynamics for different levels of image contrast.

The folder 'analyzeOptimizationStatistics' displays our analysis of fit error in the vicinity of the optimal parameter set, as explained in the new section "Robustness of fit" in the Supplementary Materials and methods.